# Helios 2.0: A Robust, Ultra-Low Power Gesture Recognition System for Event-Sensor based Wearables

## Abstract

We present an advance in machine learning powered wearable technology: a mobile-optimised, real-time, ultra-low-power gesture recognition model. This model utilizes an event camera system that enables natural hand gesture control for smart glasses. Critical challenges in hand gesture recognition include creating systems that are intuitive, adaptable to diverse users and environments, and energy-efficient allowing practical wearable applications. Our approach addresses these challenges through four key contributions: a novel machine learning model designed for ultra-low-power on device gesture recognition, a novel training methodology to improve the gesture recognition capability of the model, a novel simulator to generate synthetic micro-gesture data, and purpose-built real-world evaluation datasets. We first carefully selected microgestures: lateral thumb swipes across the index finger (in both directions) and a double pinch between thumb and index fingertips. These human-centered interactions leverage natural hand movements, ensuring intuitive usability without requiring users to learn complex command sequences. To overcome variability in users and environments, we developed a simulation methodology that enables comprehensive domain sampling without extensive real-world data collection. Our simulator synthesizes longer, multi-gesture sequences using Markov-based transitions, class-balanced sampling, and kinematic blending. We propose a sequence-based training approach to learn robust micro-gesture recognition entirely from simulated data. For energy efficiency, we introduce a five-stage, quantization-aware architecture with >99.8% of compute optimized for low-power DSP execution. We demonstrate on real-world data that our proposed model is able to generalise to challenging new users and environmental domains, achieving F1 scores above 80%. The model operates at just 6-8 mW when exploiting the Qualcomm Snapdragon Hexagon DSP. In addition, this model surpasses an F1 score of 80% in all gesture classes in user studies. This improves on the state-of-the-art for F1 accuracy by 20% with a power reduction 25x when using DSP. This advancement for the first time brings deploying ultra-low-power vision systems in wearable devices closer and opens new possibilities for seamless human-computer interaction. A real-time video demonstration of Helios 2.0 can be found at this link.

## 1 Introduction

Hand gestures represent one of the most natural human-computer interaction paradigms, offering intuitive control without requiring visual attention from the user. As the number of smart glass devices has exploded over the last year, highlighted by their emergence as a wearable product category at the Consumer Electronics Show (CES) 2025, efficient gesture interaction has become increasingly important. Currently, these devices almost exclusively use capacitive touch on the glasses' temple for their human-machine interface. This solution however is often awkward to use, limits interaction vocabulary, and can be socially conspicuous as users repeatedly reach to touch their eyewear. The industry has acknowledged these limitations by introducing additional peripherals, such as rings or wrist bands. While these accessories enable more intuitive and ergonomic hand-based input, they require users to wear and maintain an additional piece of hardware. To overcome this need for additional devices while still leveraging natural hand-based input, an event-based vision system was previously proposed by Helios 1.0 (Bhattacharyya et al., 2024).

In this work, we present significant improvements to the event-based gesture recognition approach proposed in Helios 1.0. We develop a model that is specifically designed to be ultra-low-power to enable always on smart glasses. Through this we pay attention to the device each section of the model intends to run on, ensuring that the majority of the models compute is quantized and run on a DSP device specifically optimised for this. In addition, we developed a novel simulation method that makes it possible to create large diverse training datasets without the need for expensive time consuming real-world data collection. We also develop custom training pipelines that allow us, in addition to our improved training datasets, to develop state-of-the-art gesture recognition models. This is achieved through a new method to train with multi-gesture temporal sequences, quantisation-aware training (QAT) and fine-tuning on tailored synthetic datasets. These improvements result in a model that achieves F1 accuracy greater than 70% for 2-channel models and greater than 80% for 6-channel models

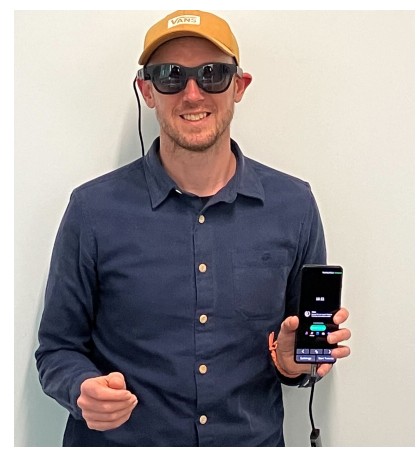

Figure 1: Helios 2.0 hardware and testing application running the machine learning model in real time.

across multiple users and environments, while only consuming 6-7 mW when running computationally intensive portions on a Qualcomm Snapdragon Hexagon DSP. The model, which is specifically designed for low-power operation, is composed of five stages, with two stages containing the majority of the computational burden running on the DSP. With a model latency of 2.34 ms, the system enables smooth and responsive interaction that feels natural to users. Figure 1 depicts a user with Helios 2.0 hardware where the proposed machine learning model runs in real time.

These advances make hand gesture interaction with smart glasses more accessible, eliminating the need for additional accessories while maintaining a socially acceptable and power-efficient interaction paradigm. The rest of the paper breaks down the specific enhancements made to the system and presents comprehensive evaluation results, before concluding with future directions for this technology.

Our key contributions within this paper are as follows:

1. A novel machine learning model specifically designed and optimised for ultra-low-power on device gesture recognition, reducing power consumption from prior work by 25x

2. A novel machine learning model that improves F1 score for gesture recognition by 20%

3. A novel training methodology to improve the model performance of event camera driven gesture recognition machine learning models, through the use of multi-gesture temporal sequences, quantisation-aware training (QAT) and fine tuning

4. A novel simulation methodology that enables training on large, diverse datasets without the need for expensive real-world data collection

5. More rigorous real-world data based testing benchmarks including increased user and environmental variability

## 2 Related Work

### 2.1 Hand Gesture Recognition

Hand Gesture Recognition (HGR) has been explored across various sensing modalities (PramodKumar & Saerbeck, 2015; Guo et al., 2021). Data gloves (Dipietro et al., 2008) provide high accuracy but are unsuitable for continuous use. Vision-based methods using traditional cameras (Zeng et al., 2018; Laskar et al., 2015; Liu et al., 2019; Bao et al., 2017; Singla et al., 2018) combined with deep learning models (Ji et al.; Koller et al., 2016; Molchanov et al., 2015; 2016; Neverova et al., 2014; Sinha et al., 2016) demonstrate strong performance in controlled environments but struggle with environmental variations and computational constraints. (Chandra & Lall, 2016) reduced frame rates and employed standby modes to save power to

achieve real-time gesture recognition. However, they faced a fundamental tradeoff: lower latency requires higher sampling rates leading to increased power consumption. Alternative approaches like sEMG (Mayor et al., 2017) and ultrasound (Xia et al., 2017) face challenges with electrode placement, muscle fatigue, or specialised hardware requirements.

**Our real-time gesture recognition contribution:** Compared to the previous state-of-the-art, our approach improves power efficiency by 25× and gesture recognition accuracy by 20%. This significantly advances the feasibility of power-efficient real-time gesture recognition for deployment in smart glasses.

## 2.2 Event Camera Datasets: Real and Synthetic

Event cameras operate on an asynchronous, per-pixel sensing mechanism, fundamentally different from conventional frame-based cameras (Gallego et al., 2022). It is inspired by the human retina where each pixel functions independently, continuously detecting changes in light intensity. More details about event cameras is provided in appendix B.1.

Real-world datasets, captured using event cameras such as DAVIS (Brandli et al., 2014) and Prophesee cover diverse applications like object detection, action recognition, tracking, and 3D perception. EventVOT (Wang et al., 2023) and eTraM (Verma et al., 2024) focus on object tracking and traffic monitoring, while DVS-Lip (Tan et al., 2022) and SeAct (Zhou et al., 2024b) explore fine-grained human motion and action recognition. Automotive datasets such as DSEC (Gehrig et al., 2021), GEN1 (de Tournemire et al., 2020), 1 MPX (Perot et al., 2020) and N-Cars (Sironi et al., 2018) provide event streams for autonomous driving. However, real-world datasets are constrained by hardware limitations, environmental conditions, and annotation challenges, making it difficult to scale data collection efficiently. To address these gaps, synthetic datasets leverage simulators to generate large-scale, controlled event streams. For example, Event-KITTI (Zhou et al., 2024a) extends the popular KITTI dataset with event-based representations. Datasets such as N-ImageNet (Kim et al., 2021), CIFAR10-DVS (Li et al., 2017), and N-MNIST (Orchard et al., 2015) convert standard image datasets into event streams, enabling model benchmarking across classification tasks. Despite these advances, synthetic datasets often lack realistic noise characteristics, which can impact generalisation to real-world scenarios.

**Our datasets contribution:** We propose the development of a specialised event-based simulator to generate microgesture data tailored to our specific application. This approach is motivated by the absence of both real-world and synthetic datasets that sufficiently address our requirements. By leveraging simulation, we can precisely control gesture variations, environmental conditions, and sensor parameters, enabling the creation of high-fidelity data that aligns with our use case. We also detail three datasets collected from internal studies, covering user variability, where environment was fixed, environment variability, with a single expert user across ambient lit complex scenes and an outdoor dataset utilising one of the groups from the user variability study. These provide benchmarks for key areas needed to access a system targeting smart-eyewear.

## 2.3 Event Camera Simulators

A review of recent event simulators can be found in (Chakravarthi et al., 2024). Notable simulators include the DAVIS Simulator (Mueggler et al., 2016), which generated event streams, intensity frames, and depth maps with high temporal precision through time interpolation. ESIM (Rebecq et al., 2018) extends this by modelling camera motion in 3D environments. Unlike traditional frame-based simulators, ESIM accurately simulates the asynchronous nature of event cameras, ensuring that events are generated only when intensity changes occur. More advanced simulators like ICNS Simulator (Joubert et al., 2021) introduce realistic noise models improving the fidelity of synthetic events, while DVS-Voltmeter (Lin et al., 2022) incorporates stochastic variations in sensor behaviour.

**Our simulation contribution:** Simulators like ESIM are not inherently designed for human-centric applications, such as gesture recognition. To address this limitation, we leverage the work of ESIM alongside our custom rendering engine, developed in Unity, to generate synthetic event data of hands performing microgestures.

### 2.4 Event-Based Vision

ML approaches for event-based classification and detection typically transform raw event data into structured representations, such as time surfaces or event volumes, which can then be processed using standard Convolutional Neural Networks (CNNs) or recurrent architectures like Conv-LSTMs (Millerdurai et al., 2024; Liang et al., 2024; Gao et al., 2024; Chen et al., 2024; Kong et al., 2024; Aliminati et al., 2024; Rebecq et al., 2019). While these methods are straightforward to implement, they introduce redundant computations and can be computationally expensive due to the dense processing of inherently sparse data. To overcome these inefficiencies, researchers have explored sparse convolutional architectures (Liu et al., 2015) which compute convolutions only at active sites with non-zero feature vectors, significantly reducing computational overhead (Peng et al., 2024; Yu et al., 2024; Zhang et al., 2024). Additionally, techniques leveraging temporal sparsity apply recursive sparse updates (Ren et al., 2024; Schaefer et al., 2022), rather than reprocessing the entire event volume. Beyond grid-based representations, graph-based methods maintain the compact, asynchronous structure of event streams, with Graph Neural Networks (GNNs) providing a framework for handling irregular and dynamic event data (Schaefer et al., 2022; Bi et al., 2019). However, this method still takes 202ms when implemented in Python and CUDA and run on Nvidia Quadro RTX. Current event-based ML methods are therefore still severely limited for on-device deployment. Quantisation techniques reduce neural network latency and power consumption by storing weights and activation tensors in lower (8-bit) precision (Nagel et al., 2021; Wu et al., 2020; Tian et al., 2024). Post-Training Quantisation (PTQ) (Shang et al., 2022) quantises pre-trained models with minimal engineering effort, while Quantisation-Aware Training (QAT) (Jacob et al., 2017) incorporates simulated quantisation during training to better preserve accuracy, especially for low-bit quantisation.

**Our machine learning contribution:** We develop a QAT-scheme for our custom architecture to significantly reduce power consumption without sacrificing gesture recognition accuracy. Helios 2.0 reduces power consumption by $25\times$ (from 150 mW to $<7$ mW), lowers latency from 60 ms to 2.4 ms, and improves F1 accuracy by 20% over Helios 1.0 (Bhattacharyya et al., 2024). Compared to Helios 1.0, Helios 2.0 uses longer training sequences spanning 2 s with six gestures per sequence (vs. one gesture per sequence in Helios 1.0), along with structured Markov-based transitions, class-balanced sampling, and kinematic motion blending for more realistic and diverse simulation data. Unlike Helios 1.0, Helios 2.0 introduces a five-stage, quantization-aware architecture with $>99.8\%$ of compute optimized for low-power DSP execution. We also redesign the training approach using longer sequences, learning gesture class probabilities from dense 900 Hz labels with thresholded aggregation. Additionally, we apply rotational augmentation during fine-tuning to improve robustness to hand pose variations. We acknowledge (Amir et al., 2017) as the closest related benchmark, but note that it involves large, deliberate gestures, making it a significantly easier and less realistic task for smart wearable interaction. In contrast, our work focuses on short, subtle microgestures suitable for real-time application control.

Our contribution fills a critical gap in benchmarking low-power, event-based wearable interfaces. Unlike Meta's approach (Kin et al., 2024), which relies on frame-based cameras and hand skeletal tracking, our method eliminates the need for explicit hand pose estimation by directly recognising gestures from raw event streams. (Amir et al., 2017) also utilises event-based gesture recognition but employs the DVS-128 camera, which is impractical for integration with smart eyewear. Additionally, we address the critical challenge of false positives from ego-motion by explicitly modeling adversarial classes alongside relevant microgestures, a concern not actively mitigated in prior works (Aliminati et al., 2024; Amir et al., 2017; Kin et al., 2024).

## 3 Hardware

### 3.1 Test Platform

Our system uses a Prophesee GenX320 event camera with a custom USB readout PCB, based on Helios 1.0

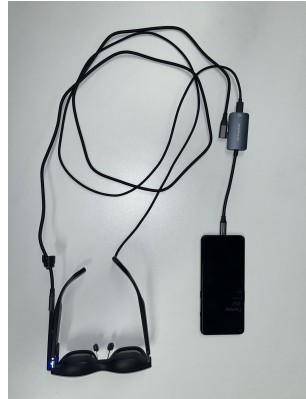

Figure 2: Hardware used for data collection and model testing. The event-camera is mounted on the left side of the glasses, with the display connection on the right.

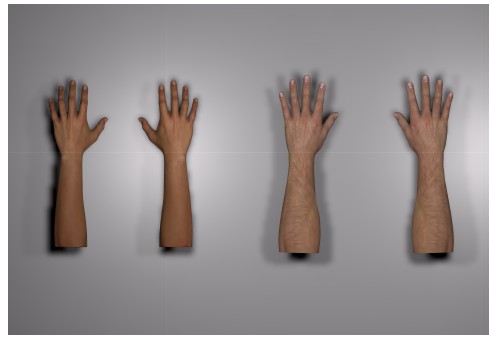

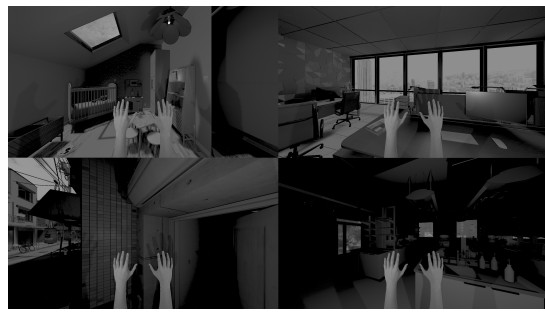

(a) 3D hand models used for simulation.          (b) 3D environments used in simulation.

Figure 3: Simulation assets used in Helios 2.0: 3D hand models and diverse 3D environments.

(Bhattacharyya et al., 2024). The key differences are where the camera was mounted and execution hardware. We mounted the event camera on XReal Airs to enable richer visual experiences and user testing. Figure 2 shows this configuration, with the event camera and USB readout on the temple, alongside the XReal's display connection. Both connect to a USB hub linked to an Android phone. For mobile-optimised model development, we used an Android OnePlus 10 Pro with the Qualcomm Snapdragon 8 Gen1, featuring Hexagon DSP for acceleration. While this offers more computing power than in Helios 1.0, it provided insights into how DSP and NPUs would enhance model performance in future smart glass products.

## 3.2 Model Power Evaluation

To determine the model power consumption a Snapdragon XR2Gen2 based platform was used. The CPU architecture is similar to the Snapdragon 8 Gen1 featuring Kryo Prime, Gold and Silver ARM based CPU cores and Hexagon DSP (Wikipedia, 2025). The power was calculated by measuring the current consumption using a 16-bit ADC that sampled the voltage across a shunt resistor in series with the device. This was multiplied by the input voltage provided by a dedicated PSU, also sampled by a 16-bit ADC, to give the total device power. To calculate the power used by the model, the following test sequence is carried out using a test harness: first the harness loads the input data for testing and runs all base tasks, except for model execution, recording the power $P_{base}$. On the next run, the harness runs and loads the same data, but this time completes $N$ iterations of the model over it. The test harness then terminates the measurement, logging the total power of the run, $P_{total}$. It should be noted the data is from a user recording, so it is representative of what the model would process at inference time. Finally, the power consumption of the model $P_{model}$ was calculated using the following $P_{model} = (P_{total} - P_{base})/N$.

# 4 Datasets

## 4.1 Synthetic Datasets

This section explains our simulation-based approach to dataset generation. By combining eSIM's event generation capabilities (Rebecq et al., 2018) with a custom Unity-powered (Unity Technologies) rendering engine, our system synthesises realistic hand gesture events for training purposes.

### 4.1.1 Simulator Details

The simulator leverages Ultraleap's *Orion* (Ultraleap Inc) hand tracking data, recorded at 90 Hz and segmented into 40-frame sequences to enable simulation-time augmentation. Pre-processing includes removing poor poses and sequences with multiple consecutive untracked frames, and filling single missing frames via linear interpolation to prevent rendering artifacts. The processed data drives hand mesh rigging (Figure 3a) within Unity's High Definition Render Pipeline. This renders photorealistic hands in synthetic 3D environments (Figure 3b).

### 4.1.2 Simulator Augmentation

Simulator augmentations fall into three categories: (1) environment modifications including lighting and hand positioning, (2) random mutation of recorded hand tracking data, and (3) formation of hand data into Helios 2.0 micro gesture classes.

Environmental augmentations include varying scene lighting brightness (50-400% of nominal values) while maintaining fixed camera exposure. Camera movement follows a path finding algorithm that navigates between 'safe' zones on the $(x, y)$ ground plane while continuously rotating around its yaw axis to simulate head motion typical of head-mounted AR glasses. For data augmentation, we reconstruct a forward kinematic model of bone angles and apply small random continuous deviations each frame, ensuring unique data generation in each simulation run. Our proposed micro gesture sequence construction is detailed in the following section.

### 4.1.3 Simulation of Longer Multiple Micro Gesture Sequences

In Helios 1.0 (Bhattacharyya et al., 2024), a 7-class model was proposed with gestures comprising (1) unknown motions, (2) untracked hands, (3) pinch, (4) double pinch, (5) swipe left, (6) swipe right, and (7) rest positions, generated in a single time window of 434 ms. In Helios 2.0, to improve system robustness, we extended sequence length from 434ms to 2s and expanded from 7 to 10 classes by adding "return" versions of several gestures. The expanded classification includes: (1) Unknown, (2) Untracked, (3) Pinch, (4) Double Pinch, (5) Pinch (return), (6) Swipe Left, (7) Swipe Left (return), (8) Swipe Right, (9) Swipe Right (return), and (10) Rest. Figure 4a illustrates swipe and pinch gestures in the simulator.

Our animation approach uses forward kinematics with motion derived from Optitrack user studies. This revealed gesture motions approximating sigmoid curves $S(x) = \dfrac{1}{1 + e^{-mx}}$, where $m$ is a hyper-parameter controlling the steepness of the sigmoid curves. We implemented four sigmoid variations with different $m$ for dataset diversity. This combined with gesture expansion explicitly addresses false positives - for example, when returning to rest position after a right swipe, the leftward motion might incorrectly trigger a left swipe, causing user frustration in menu navigation scenarios.

We redefine each gesture to be 333ms, allowing 6 different gestures within a 2s sequence. Rather than random selection, we implemented structured transition rules between gestures as a Markov chain, with rest position serving as a root state. For class balance, we manually weighted probabilities (Figure 8a) rather than using algorithmic approaches like Reverse Page Rank (Berend, 2020). Between certain transitions (rest to swipe, rest to unknown, or rest to pinch), we apply frame blending through linear interpolation in angle space to ensure kinematically plausible motions.

### 4.1.4 Rendering and Event Generation

After synthesizing sequences of hand positions, we first render frames and then use them to generate events utilising an Unity implementation of eSIM (Rebecq et al., 2018). An event is given by $P_i := (x_i, y_i, p_i, t_i)$. Here, $x_i, y_i$ are the $(x, y)$ pixel coordinates respectively, $p_i$ is the polarity defined as 1 for positive changes and 0 for negative changes and $t_i$ is the time the event occurred, in $ns$ for event $i$. To ensure contrast detection is possible across the scene, we employ High Dynamic Range (HDR) rendering to prevent event loss in regions that saturate in the 8-bit range (0-255). This accommodates more diverse environmental and lighting variations, closer to real-world scenarios. The simulator outputs events, current active gesture labels and 3D hand joint location at 900Hz (rather than the input data's 90Hz) for precise event timing.

To validate our simulator, a dynamic target was created both in the lab and in our simulator. The sim-to-real gap in event rate was first tuned on a high-contrast target. It was then fine-tuned with real-world hand gesture data to the final contrast ratio.

### 4.2 Real Dataset Curation

We developed an Android testing application to collect real world gesture data for model benchmarking. The app captured user experience level, personal characteristics (hand size, skin tone using the Monk Skin Tone Scale), and environmental factors (lighting, motion status). The tests consisted of 60 trials where users performed random prompted gestures (pinch, left/right swipe) without visual feedback to prevent behaviour

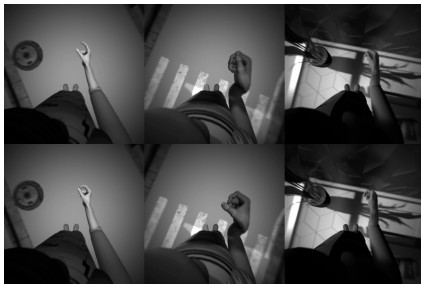 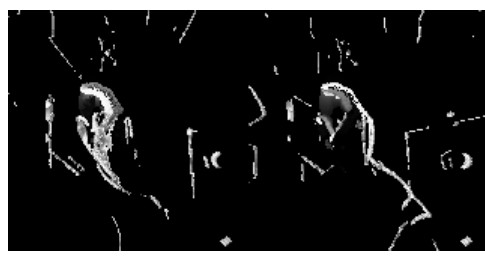

(a) Left to right: pinch, left swipe, right swipe. The end point of the microgesture is illustrated at the bottom.

(b) Example time surface with left hand side illustrating the positive polarity image and right hand side illustrating the negative polarity image.

Figure 4: Gestures modeled for smart interactions by Helios 2.0 and an example of time surface representation.

modification. Each trial logged the requested gesture, timing information, and raw event stream data for subsequent analysis.

The datasets created using internal users utilised the Monk Skin Tone Scale (Monk, 2023). An illustration of the scale is shown in Figure 9 in the supplementary. Using this scale enables a qualitative way to assess coverage of benchmark datasets. This ensures that as we expand the datasets we ensure equal representation.

### 4.2.1 Human Variability Study

In the Human Variability Dataset, the user was asked to stand within a square and were oriented using an arrow marked on the floor. This ensured that between users the background did not change. An RGB image from the view point of the event camera is shown in Figure 10 in the supplementary, illustrating the background used. Twenty users with varying experience levels completed the testing protocol under controlled office conditions. These twenty users were divided into four groups for systematic data collection. In Figure 8b in the supplementary we show the distribution of hand sizes (measured from the base of the wrist to tip of the middle finger) and skin tones among participants. Video recordings supplemented quantitative metrics to qualitatively analyse low-performing cases.

### 4.2.2 Scene Variability Study

To test across more challenging environments, the Scene Variability Dataset was created. This consists of different floor textures, with varying light levels. An RGB image is shown in Figure 11 in the supplementary, illustrating the variation in backgrounds used. An expert user conducted test sequences across four different home environments with only natural lighting. These included various floor textures, with light levels between 8-240 lux to test the model robustness.

### 4.2.3 Outdoor Performance Study

In the final dataset we consider an outdoor scene. An RGB reference image is shown in Figure 12 in the supplementary. Group 2 participants from the human variability study (from section 4.2.1 ) repeated the same test sequence outdoors. Tests occurred within a 30-minute window with consistent light levels of 3000 lux.

## 5 Methodology

### 5.1 Machine Learning Model

We developed a mobile-friendly machine learning architecture for microgesture detection optimised for low-power, low-latency performance on both CPU and DSP targets. This makes it particularly well-suited to always-on smart eyewear.

### 5.1.1 Event Representation for Helios 2.0

In this section, we first describe how the event stream generated in section 4.1.4 is processed before being used to train models. We propose to use polarity-separated event Time Surfaces (TS) as our event representation for training. We build on the Locally-Normalised Event Surfaces (LNES) approach from (Rudnev et al., 2021). This achieves both computational and storage efficiency. By separating polarities, we also mitigate event clashes and ensure a more structured and accurate representation of the temporal dynamics in event data. A key technical challenge is integrating the TS representation with the synthetic dataset produced in section 4.1.4.

To construct the TS representation $\mathcal{I}_k \in \mathbb{R}^{w \times h \times 2}$, we divide an event stream of length $L$ into discrete time windows. In our longer-sequence datasets, we set $L = 2\,\text{s}$, which is split into 25 time steps of 80 ms each. Thus, $k \in \{0, 1, \ldots, 24\}$ indexes each of these 25 windows. We denote the duration of each time window by $T_s$. $\mathcal{I}_k$ is created by first initialising it with zeros and collecting events that have incoming timestamps $T_i$ within this window by iterating through from the oldest to the newest event. We reduced $T_s$ from 434ms to 240ms to improve temporal consistency for shorter gestures in longer windows. To create LNES, we initialise the TS $\mathcal{I}_k$ to zeros and update pixel values with an exponential decay:

$$\mathcal{I}_k = \begin{cases} e^{-\lambda \left( \dfrac{T_{max} - T_i}{T_s} \right)}, & \text{if } T_{max} - T_i < T_s \\ 0, & \text{otherwise} \end{cases} \tag{1}$$

Here $T_{max}$ is the maximum timestamp in the window, $\lambda = 5$ is the decay constant, and $T_i$ is the timestamp of the incoming event. During inference, pixels with timestamps older than $T_s$ are reset to zero.

For an event stream with image bounds $\{w, h\}$, we accumulate positive polarity events in the range $\{w, h\}$ and negative polarity events in the range $\{w, 2h\}$, thereby reshaping $\mathcal{I}_k \in \mathbb{R}^{w \times 2h}$ (fig. 4b - note for illustration purposes Figure 4b displays the image as $\{2w, h\}$). The 3D hand joint locations are reprojected into 2D camera space to obtain bounding box locations. Points beyond image bounds are clipped to $\{[0, w-1], [0, h-1]\}$. Bounding box corners are computed via min-max operations, excluding points below the wrist for better box tightness. As an improvement on prior work, we enforce square bounding boxes for improved training accuracy and reduced sim-to-real gap at inference time.

### 5.1.2 Helios 2.0 Model Architecture and Quantisation

Helios 2.0 introduces a five-stage, quantisation-aware architecture with >99.8% of compute optimized for low-power DSP execution (Figure 5).

- Stage 1: Downsamples the input time surface (TS) representation to a lower dimensional representation.

- Stage 2: Comprises of convolutional layers and dense layers that perform feature extraction. The number of layers is specifically chosen to most efficiently extract the features by initially lowering the spatial resolution before predicting dense features. This layer contains a significant amount of the models parameters and is quantized to run efficiently on a DSP.

- Stage 3: Has a non-quantized dense layer to predict floating point values for a bounding box, which ensures the model can predict the bounding box with high accuracy. Then stage 3 crops and resizes the original input using the predicted bounding box.

- Stage 4: Comprises of convolutional layers and dense layers that perform feature extraction on the crop of the hand within the original image. This is again specifically chosen to most efficiently extract the features by initially lowering the spatial resolution before predicting dense features. This layer in addition to stage 2 makes up the majority of the models parameters and is quantized to run efficiently on a DSP.

- Stage 5: Has a single dense layer to predict the micro gesture that is contained within the cropped image. It then combines stage 4 predictions with stage 2 hand-presence probabilities to produce final microgesture predictions.

Figure 5: High-level block diagram of the five stage microgesture detection model architecture

The CNN components in stages 2 and 4 account for >99.8% of parameters and floating point operations. We quantised these stages to run on a Qualcomm Hexagon DSP, reducing power consumption and inference time. The final dense layers of these stages and the remaining three stages (which include interpolation and softmax functions) use 32-bit floating-point computation to maintain accuracy, as their minimal compute requirements make quantisation unnecessary.

We implemented quantisation-aware training (QAT) with 8-bit symmetric, per-channel weight quantisation and 8-bit asymmetric, per-tensor activation quantisation. This scheme is natively supported by Hexagon DSP hardware that balances accuracy, latency, and power consumption. QAT is particularly important for our parameter-constrained model to mitigate quantisation noise effects.

### 5.1.3 Training Helios 2.0 with Proposed Simulated Dataset

A key contribution of our work is training the model proposed in section 5.1.2 with the longer sequence dataset proposed in section 4.1.4 and their TS representations proposed in section 5.1.1. We aim to learn the conditional probability $p(g|\mathcal{I}_k)$ of predicting gesture class $g$ given input $\mathcal{I}_k$, where $\mathcal{I}_k$ is the TS representation for time-step $k$ produced in section 5.1.1. Our datasets feature six distinct gestures within a 2 s sequence, each lasting $\sim$333 ms.

As outlined in section 4.1.4 and section 5.1.1, the simulator generates gesture labels at 900 Hz. This yields 216 labels per 240 ms window. To obtain $g$, the labels must be aggregated. In the case of the first sample in a sequence, we select the maximum across all 216 labels as $g$. Subsequently, we assign a new label only if its proportion exceeds the gesture threshold (e.g., at 0.6 threshold, a transition from 'rest' to 'swipe right' requires at least 60% of labels to be 'swipe right'). This results in a corresponding $g$ for each $\mathcal{I}_k$ generated. We refer to these models as the 2-channels models.

To capture broader temporal context, we feed three consecutive TS samples ($\mathcal{I}_k$, $\mathcal{I}_{k-1}$, and $\mathcal{I}_{k-2}$) into the model, effectively capturing gestural evolution (e.g., a 'swipe left' should be followed by a 'swipe left return'). Including positive and negative polarities, this representation has dimensionality {w, 6h}. We refer to these models as the 6-channels models.

We provide more details about our training and inference setup in appendix D.

### 5.2 Finetuning Helios 2.0

To address the lack of rotation invariance in CNN features, we augment our training dataset with rotational variations and create an augmented finetuning dataset. While the longer-sequence dataset (section 4.1.3) captures diverse microgestures, it lacks comprehensive hand pose and orientation coverage. We apply uniform random rotations (25°- 40°) to each 2 s sequence, enhancing the model's robustness to rotational variance. During fine-tuning, we freeze stages 1-3 and train only stages 4-5 with a reduced learning rate. This preserves bounding-box localisation and low-level features while adapting to rotational variations.

## 6 Experiments

In this section, we present results from our quantitative experiments. We first evaluate our 2-channel low-power QAT model trained from scratch on simulated data (90% training, 10% validation) in section 6.1. We render two versions of our longer sequences datasets (described in section 4.1.4): 1× and 4×. 1× has $\sim$25$k$ training samples per microgesture class, while 4× has $\sim$100$k$. We also apply the rotation augmentation as described in section 5.2 on our 4× dataset to create a 4× Augmented dataset.

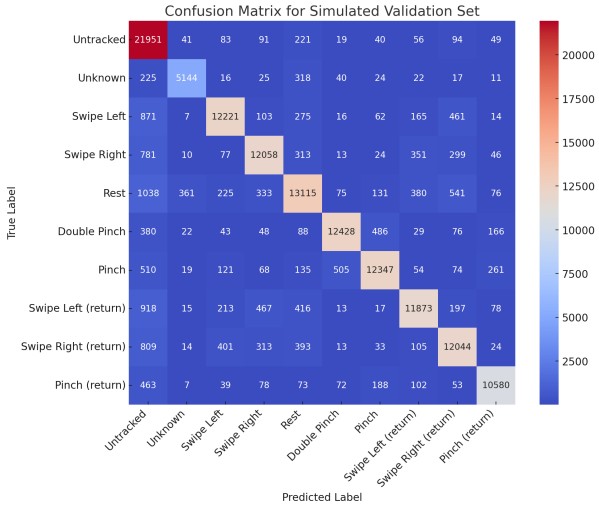

Figure 6: Confusion matrix for the 2-channels TS QAT model trained from scratch on the simulated validation set across 10 proposed classes.

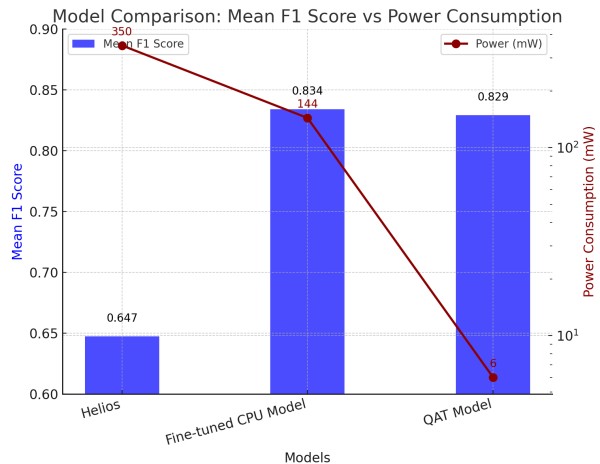

Figure 7: Mean F1 score (across RS, LS and CP) for user averages plotted against power consumption for Helios 1.0 (Bhattacharyya et al., 2024), our fine-tuned CPU model, and our QAT model.

In section 6.2, we then assess both 2-channels and 6-channels models across both CPU and DSP targets on real-world office environment human variability data (section 4.2.1). Next, we evaluate the QAT models on outdoor datasets (section 4.2.3) and scene-variability (section 4.2.2) in section 6.3 and section 6.4 respectively.

After that, we describe results covering QAT training schemes (section 6.5) Finally, we present an ablation study of key model components in low-power quantized architectures in section 6.6.

In the appendix, we provide more extensive model ablations (appendix E) and our proposed model performance generalisability beyond a single training instance (appendix F).

Using ground truth labels, we quantified true-positive (TP), false-positive (FP), and false-negative (FN) to calculate mean and median F1 scores across microgesture classes, alongside power consumption and latency measurements for both model configurations. For our experimental evaluation on real datasets, we report results on three microgesture classes - swipe right (RS), swipe left (LS) and combined pinch (CP, which is a combination of pinch and double-pinch classes).

## 6.1 Performance on Simulated Data

To evaluate our 2-channel low-power QAT model trained on simulated data, we analysed its confusion matrix across all 10 gesture classes (Figure 6). The matrix's diagonal dominance shows effective gesture discrimination, with minimal confusion between kinematically similar gestures (e.g., 2.8% between 'swipe right (return)' and 'left swipe'). The model achieved 87.9% average class-wise precision, demonstrating successful feature learning despite quantisation constraints and readiness for real-world deployment.

## 6.2 User Testing for Human Variability

In this section, we present results on real-world test data collected as described in section 4.2.1 to evaluate our model's performance to human variability. In Figure 7, we compare our Helios 2.0 CPU and QAT models against the Helios 1.0 model presented in (Bhattacharyya et al., 2024). Given Helios 1.0 is a CPU only model, comparing it our Helios 2.0 CPU model provides the most like-for-like comparison. This shows that our model improves the F1 score from 0.647 to 0.834 as well as reducing the power requirement from 350mW to 144mW. It should be noted however that the CPU and memory architecture are different.

As we designed our Helios 2.0 model such that two of the five stages (two and four) can be run on DSP we also compare to the QAT model. By quantizing those stages the F1 score reduces from 0.834 to 0.829, which

Table 1: Comparison of Model Performance Across Helios 1.0 (Bhattacharyya et al., 2024) and our proposed Helios 2.0 2-channels and 6-channels models. All models are trained for 10 Epochs. Here we primarily evaluate performance for microgestures: RS stands for right swipe, LS is left swipe and CP is combined pinch. This is evaluated on the Human Variability Study Dataset detailed in section 4.2.1.

| Model | Type | Training Data | Mean F1 RS | Mean F1 LS | Mean F1 CP | Median F1 RS | Median F1 LS | Median F1 CP | Power (mW) | Latency (ms) |
|---|---|---|---|---|---|---|---|---|---|---|
| Helios 1.0 | CPU Model | – | 0.6336 | 0.7314 | 0.5773 | 0.6826 | 0.8667 | 0.5965 | 350 | 60 |
| 2-channels TS | CPU Model | 4× | 0.7382 | 0.5229 | 0.7552 | 0.7214 | 0.5714 | 0.8568 | 144 | 6.15 |
| | Fine-tuned CPU | 4× Augmented | 0.8081 | 0.8364 | 0.8572 | 0.9023 | 0.9070 | 0.9505 | 144 | 6.15 |
| | QAT Model | 4× Augmented | 0.8209 | 0.8244 | 0.8421 | 0.9045 | 0.8797 | 0.9500 | 6 | 2.35 |
| 6-channels TS | CPU Model | 4× | 0.7780 | 0.5477 | 0.7681 | 0.7625 | 0.6591 | 0.8739 | 172 | 7.46 |
| | Fine-tuned CPU | 4× Augmented | 0.8512 | 0.8363 | 0.8429 | 0.9316 | 0.9115 | 0.9249 | 172 | 7.46 |
| | QAT Model | 4× Augmented | 0.8149 | 0.8362 | 0.8902 | 0.9256 | 0.8940 | 0.9500 | 8 | 4.60 |

Table 2: Comparison of F1 Scores Across 2-channels and 6-channels QAT Models and 3 Real Datasets: Human Variability (section 4.2.1), Outdoors (section 4.2.3 and Scene Variability (section 4.2.2).

| Model | Real Dataset | Mean F1 | | | Median F1 | | |
|---|---|---|---|---|---|---|---|
| | | RS | LS | CP | RS | LS | CP |
| 2-channels | Human Variability (Group 2) | 0.7365 | 0.7517 | 0.5439 | 0.8856 | 0.8558 | 0.4259 |
| 6-channels | Human Variability (Group 2) | 0.7806 | 0.7540 | 0.7100 | 0.9180 | 0.8576 | 0.6750 |
| 2-channels | Outdoors (Group 2) | 0.8815 | **0.8440** | 0.7084 | 0.8980 | **0.9189** | 0.8980 |
| 6-channels | Outdoors (Group 2) | **0.9468** | 0.8309 | **0.8375** | **0.9499** | 0.8743 | **0.9744** |
| 2-channels | Human Variability (User 3) | **0.9744** | **0.9756** | **1.0000** | **0.9744** | **0.9756** | **1.0000** |
| 6-channels | Human Variability (User 3) | **0.9744** | **0.9756** | **1.0000** | **0.9744** | **0.9756** | **1.0000** |
| 2-channels | Scene Variability (User 3) | 0.8702 | 0.7496 | 0.6962 | 0.8889 | 0.7242 | 0.7097 |
| 6-channels | Scene Variability (User 3) | 0.8789 | 0.8436 | 0.8363 | 0.9041 | 0.8998 | 0.9189 |

is a minor reduction in performance, whilst reducing the power to 6mW. This means our model improves F1 score by 30% while only requiring 2.3% of the power.

## 6.3 User Testing for Environmental Variability

Performance comparison between Human Variability (section 4.2.1) and Outdoor datasets (section 4.2.3) shown in Table 2 reveals consistent improvements in outdoor environments. The 2-channel TS model shows notable gains across all gestures outdoors, with Right Swipe improving from 0.7365 to 0.8815 (mean F1) and Combined-Pinch dramatically improving from 0.4259 to 0.898 (median F1). Similarly, the 6-channel TS model demonstrates enhanced outdoor performance, particularly for Right Swipe (0.7806 to 0.9468 mean F1) and Combined-Pinch (0.71 to 0.8375 mean F1, with median F1 jumping from 0.675 to 0.9744). These improvements show our quantised models not only maintain effectiveness in outdoor conditions but often perform better than controlled environments. This is likely due to stronger gesture signals against natural backgrounds and reduction in noise in brighter conditions.

## 6.4 User Testing for Scene Variability

Our analysis reveals a stark contrast between simple indoor office environments as used in the Human Variability Dataset (section 4.2.1) and more complex environments used in the Scene Variability Dataset (section 4.2.2), as shown in Table 2. Both 2-channels and 6-channels models achieved identical, near-perfect scores in an office based environment (F1 scores of 0.97 for Right/Left Swipe and 1.0 for Combined-Pinch). However, introducing scene variability caused noticeable performance degradation, particularly for the 2-channel model, where Combined-Pinch recognition dropped from perfect scores to 0.6962 (mean F1). The

Table 3: Impact of different training strategies for Quantisation-aware training (QAT). Training options include: training QAT from scratch; training a base model and then QAT; and training a base model, finetuning it and then QAT. This is evaluated on the Human Variability Study Dataset detailed in section 4.2.1.

| QAT Model Training Strategy | Training Stage | Epochs | Dataset | $F1_{RS}$ | $F1_{LS}$ | $F1_{CP}$ | Med-$F1_{RS}$ | Med-$F1_{LS}$ | Med-$F1_{CP}$ |
|---|---|---|---|---|---|---|---|---|---|
| From scratch | QAT | 10 | 4× Augmented | **0.8209** | 0.8244 | 0.8421 | 0.9045 | 0.8797 | 0.95 |
| From 1× Base Model | Base
QAT | 10
5 | 1×
4× Augmented | 0.7771 | 0.8127 | **0.8638** | 0.9237 | 0.8896 | 0.9629 |
| From 4× Fine-tuned 1× Base Model | Base
Fine-tuned
QAT | 10
5
5 | 1×
4× Augmented
4× Augmented | 0.7699 | **0.8342** | 0.8496 | 0.906 | **0.9034** | 0.9744 |
| From 4× Fine-tuned 4× Base Model | Base
Fine-tuned
QAT | 10
5
5 | 4×
4× Augmented
4× Augmented | 0.8189 | 0.8033 | 0.8571 | **0.9379** | 0.8782 | **1.0000** |

6-channels model demonstrated superior robustness to changing backgrounds, maintaining higher performance across all gestures and showing particularly significant advantages for Left Swipe (0.8436 vs. 0.7496 mean F1) and Combined-Pinch (0.8363 vs. 0.6962 mean F1). These results underscore the importance of background complexity in the training data. It also demonstrates that additional temporal information can help in difficult conditions.

## 6.5 QAT Training Strategies

We can set up QAT in different ways: (a) training a model from scratch (b) training from a base model and then applying QAT (c) training a base model, fine-tuning on augmented data and then applying QAT. Table 3 shows the impact of different training strategies.

Based on the mean F1 scores across all three metrics we find that the training from scratch and from training 4x base model + 4x fine-tuned perform the strongest. Both of these models scoring above 0.8 F1 score across all micro-gestures. When considering the median scores, the model that has been trained from a 4x base model and then fine-tuned on 4x data achieves the highest average F1 score across all three micro-gestures.

## 6.6 Ablation Studies

Our ablation study (Table 1) shows that the TS models of 2-channels and 6-channels benefit from fine-tuning with rotation augmentation. This improvement occurs because the pre-training dataset, while useful for learning multiple micro-gesture sequences, lacks natural hand orientations. Fine-tuning helps the model adapt to realistic human hand distributions. While 6-channels TS input yields modest performance gains, it requires triple the data throughput, potentially increasing system power demands.

## 7 Conclusion

In this paper, we propose Helios 2.0, a novel, ultra-low-power event-based vision system for intuitive hand gesture control in smart glasses. This addresses key challenges in power efficiency, adaptability, and user experience. Our system leverages a minimal set of microgestures that align with natural hand movements, making interactions more intuitive. By introducing an enhanced simulation methodology, a power-optimized model architecture, and comprehensive benchmarking, we achieve state-of-the-art F1 accuracy while operating at just 6-8 mW on a Qualcomm Snapdragon Hexagon DSP. These advancements significantly improve upon prior work, surpassing existing F1 scores by 20% while reducing power consumption by 25x. Our results are the first to comprehensively demonstrate that event-based vision can provide accurate, low-power, low-latency gesture recognition, making it an ideal solution for next-generation smart glasses. In future work, we want to expand the gesture vocabulary, improve adaptation to individual user preferences, and integrating multi-modal sensing for more robust interactions. As smart glasses continue to gain traction, our approach represents a critical step toward seamless, touch free human-computer interaction in wearable technology.

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

## A    Appendix

## B    Extension of Related Work

The following sections provide an overview of event cameras and the representations commonly used for event-based machine learning applications.

### B.1    Overview of Event Cameras

Event sensors offer unprecedented temporal resolution while operating at extremely low power levels, as low as 3 mW. In contrast, traditional frame-based sensors require a trade-off between temporal resolution and power consumption, typically ranging from 35 to 200 mW, depending on the frame rate. Event cameras operate on an asynchronous, per-pixel sensing mechanism, fundamentally different from conventional frame-based cameras (Gallego et al., 2022). It is inspired by the human retina where each pixel functions independently, continuously detecting changes in light intensity.

When the intensity at a event camera pixel surpasses a predefined threshold, the sensor records an event in the form of $\langle x, y, t, p \rangle$, where $(x, y)$ represent the pixel coordinates, $t$ is the event timestamp, and $p \in \{1, -1\}$ indicates whether the light intensity increased or decreased.

While conventional machine vision architectures are optimised for frame-based data, they struggle to meet the demands of wearable and low-power applications, driving the need for event-based vision research. This section provides an overview of the current landscape of event camera datasets and simulators, which are essential for developing and training event-driven machine learning algorithms.

### B.2    Event Representations

Two widely used representations of event data for machine learning applications are time surfaces and event volumes (Gallego et al., 2022). Time surfaces store the timestamp of the most recent event for each pixel and polarity, forming a 2D map where each pixel retains a single time value. This compact representation efficiently encodes temporal dynamics but becomes less effective in textured scenes where pixels generate frequent events. By aggregating local memory time surfaces, Histograms of Averaged Time Surfaces (HATS) (Sironi et al., 2018) construct a higher-order representation to improve temporal and noise robustness.

In contrast, event volumes represent events as 3D histograms, maintaining richer temporal structure by accumulating events over time. While event volumes provide a more detailed temporal representation, they may lose polarity information due to voxel-based accumulation, potentially reducing their effectiveness in tasks requiring fine-grained contrast changes.

In our work, we employ polarity-separated time surfaces (Rudnev et al., 2021) as our event representation to achieve both computational and storage efficiency. By separating polarities, we mitigate event clashes and ensure a more structured and accurate representation of the temporal dynamics in event data.

## C    Additional Details for Dataset Creation

Here we first show the distribution of the classes in our simulated training dataset in Figure 8a.

In addition, we provide some visual examples of each of the real datasets used when analysing our models performance. For the human variability study, an RGB image from the view point of the event camera is shown in Figure 10. For the scene variability study, an RGB image is shown in Figure 11. For the outdoor dataset study, an RGB reference image is shown in Figure 12. We also show the skin tone scale used throughout in Figure 9. Finally, we present histograms of hand sizes used in our experiments in Figure 8b.

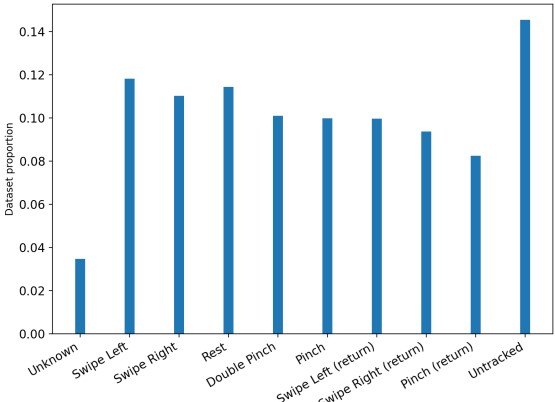

(a) Distribution of classes from a subset of the simu-
lated training data.

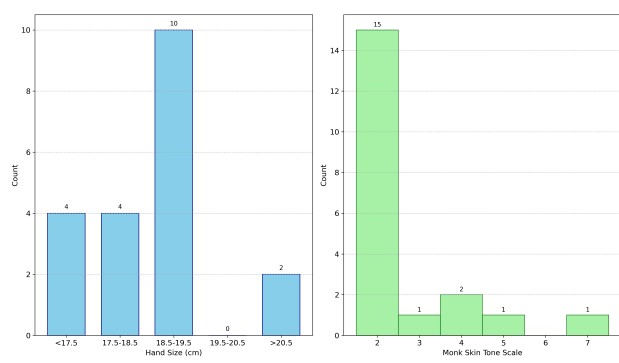

(b) Histograms of user details. Left shows distribution of
hand size and right shows distribution of skin tone based
on the Monk Skin Tone Scale (Monk, 2023).

Figure 8: Distribution of gestures in the training data, and a histogram of the user hand sizes in the testing
data for the human variability study.

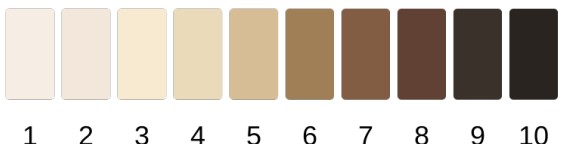

Figure 9: The Monk Skin Tone Scale, with corresponding numbering from 1 light to 10 dark.

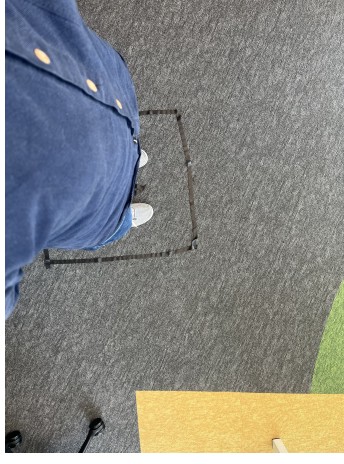

Figure 10: Image showing background used for user studies, along with floor marking to ensure minimal
background variability between users. Lighting was measured at 60 lux.

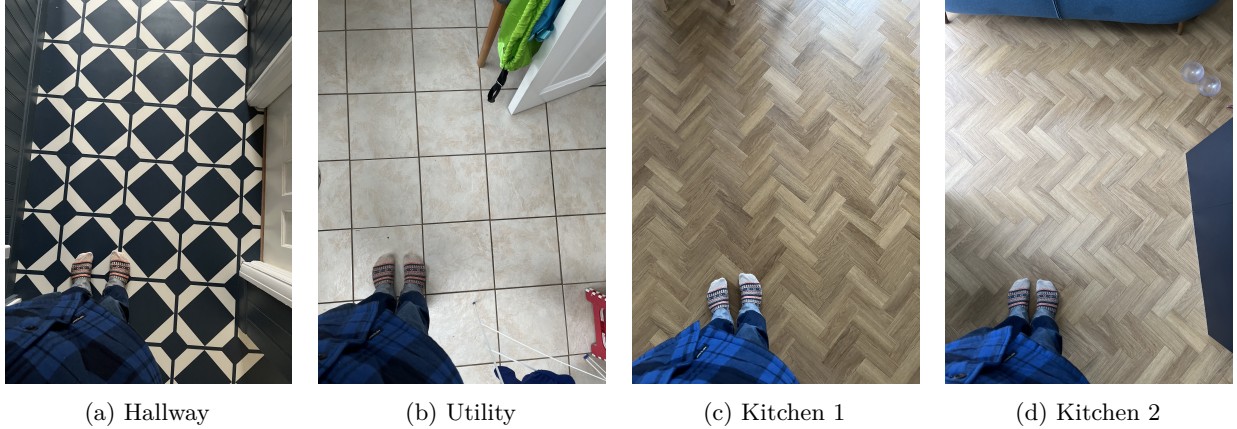

(a) Hallway          (b) Utility          (c) Kitchen 1          (d) Kitchen 2

Figure 11: Various backgrounds with natural illumination measured at the event camera position: (a) 8 lux, (b) 140 lux, (c) 30 lux, (d) 240 lux.

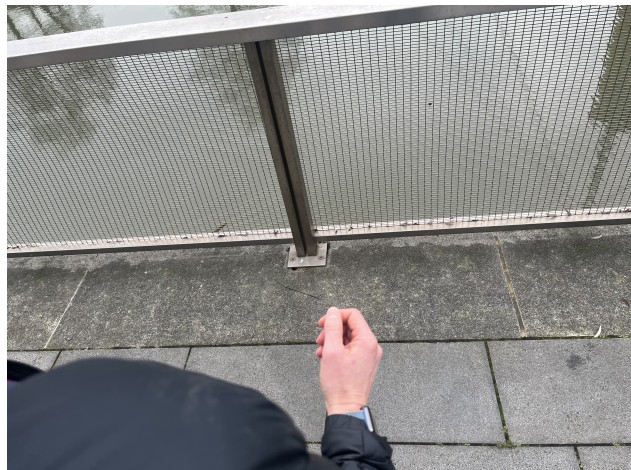

Figure 12: Outdoor scene showing hand in foreground with paving stones as a background. Ambient light= 3000 lux.

# D   Training Parametrisation and Inference Details

We trained the network for 10 epochs with a batch size of 512. We use the Adam (Kingma & Ba, 2015) optimiser with an initial learning rate of 0.0005. We utilise a learning rate scheduler, which linearly decays the learning rate after an initial period where the learning rate is held fixed. We apply dropout with 0.2 drop rate on the dense layers and set the gesture threshold value to 0.6.

At inference time the model takes as input TS of window size 240 ms. To avoid sampling issues we follow the approach of Helios 1.0 (Bhattacharyya et al., 2024) and create an TS every 80 ms. This follows the intuition that the model will be less confident on predictions where non-complete microgestures have occurred in the input frame. The model will therefore predict a microgesture with a larger probability when a complete microgesture has occurred and a lower probability when a non-complete microgesture has occurred. We also overcome the issue of the model predicting a microgesture with low confidence by introducing a threshold on softmax probabilities, at which a microgesture is deemed to have occurred, and set it to 0.65.

## D.1   Loss Function

Following Helios 1.0 (Bhattacharyya et al., 2024), we employ two loss functions: $\mathcal{L}_{\text{bbox}}$, which calculates mean squared error between predicted and true bounding boxes (only when hands are present), and $\mathcal{L}_{\text{gesture}}$, a sparse categorical cross entropy loss for microgesture classification. $\mathcal{L}_{\text{bbox}}$ is essential because $\mathcal{L}_{\text{gesture}}$ alone cannot effectively guide hand centring in TS crops, and without it, the model initially focuses solely on distinguishing the no-hand class before learning other microgestures, leading to suboptimal performance due to local minima.

# E   Model Ablation

Table 4 compares various model modifications. Adding extra dense layers or replacing them with conv1D produced mixed results with minimal impact, suggesting small model capacity increases and layer type have limited effect on performance. Replacing conv2D with depthwise separable conv2D reduced FLOPs but showed inconsistent results, with decreasing mean F1 scores while slightly improving median scores. This indicates better performance for already well-served users at the expense of others. Other configurations (strided downsampling, depthwise separable conv2D with conv1D, global average pooling, and larger conv1D strides) all reduced both model size and performance, indicating a minimum capacity threshold required for effective micro-gesture learning.

Table 4: Ablation of quantisation-aware training (QAT) model components. This ablates different architectural choices like Conv2D, Depthwise separable Conv2D, Dense layers, Conv1D, Global Average Pooling (GAP) and strided-downsampling in different stages of our 5-stage proposed model. We use the 2-channels QAT model (described in section 5.1.2) as our control model. This is evaluated on the Human Variability Study Dataset (section 4.2.1).

| Model | Params, FLOPs | Mean F1 | | | Median F1 | | | Power (mW) |
|---|---|---|---|---|---|---|---|---|
| | | RS | LS | CP | RS | LS | CP | |
| 2-channels TS (control) | 596k, 164M | 0.8209 | **0.8244** | 0.8421 | 0.9045 | **0.8797** | 0.9500 | 6 |
| Add 2 Dense layers in Stage 5 | 604k, 164M | 0.7997 | 0.8032 | 0.8559 | 0.9094 | 0.8621 | 0.9360 | 5 |
| Conv1D in place of Dense in Stages 2&4 | 605k, 165M | **0.8232** | 0.8115 | **0.8675** | **0.9279** | 0.8752 | 0.9500 | 5 |
| Depthwise Separable Conv2D in Stages 2&4 | 1.2M, 55M | 0.7721 | 0.7988 | 0.7902 | 0.8971 | 0.8842 | **0.9744** | 6 |
| Depthwise Separable Conv2D + strided downsampling in Stages 2&4 | 391k, 25M | 0.7624 | 0.6224 | 0.7365 | 0.8489 | 0.6559 | 0.9175 | **4** |
| Depthwise Separable Conv2D + Conv1D (Dense→Conv1D) in Stages 2&4 | **113k, 24M** | 0.7540 | 0.7817 | 0.7496 | 0.8514 | 0.8571 | 0.8990 | **4** |
| Depthwise Separable Conv2D + GAP (Dense→GAP) in Stages 2&4 | 70k, 52M | 0.4054 | 0.2267 | 0.5466 | 0.2942 | 0.1765 | 0.6579 | **4** |
| Depthwise Separable Conv2D + Conv1D (stride=4) in Stages 2&4 | 86k,53M | 0.6410 | 0.7003 | 0.7446 | 0.6667 | 0.8090 | 0.8932 | **4** |

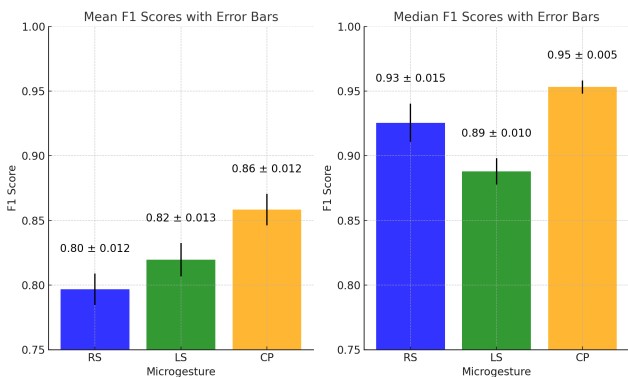

Figure 13: Mean and Median F1 Scores with Error Bars across 5 seeds for microgestures for training 2-channels TS QAT models.

## F  QAT Model Performance and Seed Variability

We analyse seed variability using standard deviation $\sigma$ as an error measure in Figure 13. We compute $\sigma$ across five independent runs for both mean and median F1 scores. Seed variability for QAT models is a crucial factor in assessing its generalisability beyond a single training instance. We observe that mean F1 scores exhibit larger error bars. In contrast, median F1 scores have smaller error bars, suggesting that the model performs consistently well in most runs. CP exhibits the lowest median variability ($\sigma = 0.005$), making it the most stable gesture. LS follows with slightly higher median variability ($\sigma = 0.010$), while RS demonstrates the highest fluctuation ($\sigma = 0.015$). The contrast between CP and RS highlights a difference in recognition stability - the robust classification performance of CP remains stable throughout the tests, while RS exhibits greater sensitivity to specific seed initialisations.

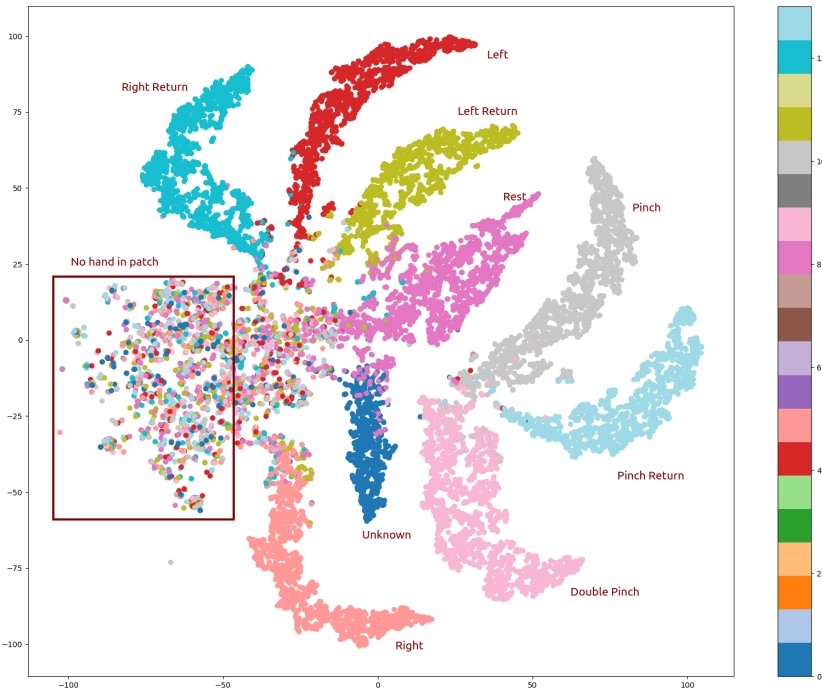

Figure 14: TSNE for 2-channels TS Fine-tuned CPU model across 10 classes

## G   Helios 2.0 t-SNE Visualisation of Learned Feature Embeddings

Figure 14 illustrates the t-SNE visualisation of our 2-channels QAT model's learned feature embeddings from the dense layer in Stage 5, across the 10 microgesture classes. The visualisation reveals distinct clustering patterns for each gesture type, demonstrating the model's ability to learn discriminative representations despite the quantisation constraints. Interestingly, directional gestures such as 'Left', 'Right', 'Left Return', and 'Right Return' form well-separated clusters with clear trajectory patterns. Also, 'Pinch', 'Double Pinch', and 'Pinch Return' classes show structured separation with minimal overlap. The 'Rest' class manifests as a cluster with moderate dispersion. Particularly interesting is the 'No hand in patch' region (highlighted by the red box), which shows a mixed distribution of points, indicating the model's uncertainty when no clear hand features are present. The 'Unknown' class forms a concentrated cluster, suggesting consistent identification of ambiguous hand motions. This embedding structure provides visual evidence of the model's strong feature extraction capabilities, which directly contributes to the classification performance observed in our quantitative evaluations.

