# OpenReview forum: "Helios 2.0: A Robust, Ultra-Low Power Gesture Recognition System for Event-Sensor based Wearables"
_TMLR — Rejected by TMLR_

### Review · Reviewer_SNtM · 2025-09-03

**Summary Of Contributions:**

This paper present a wearable system with an event camera for low power gesture recognition. The authors propose to use specific gestures for recognition and build up a simulated environment to capture enough amount of data. The authors then use a quantization-aware convolutional network for this detection task. The proposed method is evaluated on real-world data from variable setups.

The strength of this paper:
1. The method is power efficient and combines the best side of event camera. The model design is efficient and low latency.
2. Using synthetic data makes a lot of sense with this setup as event camera does not strongly depend on the textures of the input.

Weakness:
1. The proposed method is not well compared with previous method. There is no baseline method to help understand the effectiveness of the proposed method.
2. The reader is not sure if TMLR is the best venue for publicity of this paper. This paper focuses more on real-world system which may be more interesting to edge computing or embedding system etc. The contribution on learning side is limited as the convolutional network architecture have been pretty common.

In conclusion, the reader believes the weakness outweighs its strength given the limited quantitive comparisons with other methods.

The reviewer did **not** receive response from the authors therefore would keep the recommendation of **rejecting** the submission.

**Audience:**

Yes

**Audience Explanation:**

Ultra low power event camera system with real-world application is worth sharing to the community members on multiple fields such as low level vision, neuralmorphic computing etc.

**Broader Impact Concerns:**

The reviewer did not observe immediate impact concerns as a system for gesture control.

**Claims And Evidence:**

No

**Claims Explanation:**

Some of the claims are not clearly supported by experimental evidence. For example, the authors mentioned they carefully selected microgestures. However, no experimental data is given for the selection. Besides, the experimental results lack of baseline method to understand the accuracy of a general machine learning model or performance on a novel method. Gesture recognition with event camera is not a completely new problem with works like [1] and others. Even if there is no method can be compared side by side, it's still beneficial to have RGB based method for comparison.


[1] Amir, Arnon, Brian Taba, David Berg, Timothy Melano, Jeffrey McKinstry, Carmelo Di Nolfo, Tapan Nayak et al. "A low power, fully event-based gesture recognition system." In Proceedings of the IEEE conference on computer vision and pattern recognition, pp. 7243-7252. 2017.

**Requested Changes:**

The following improvements would simply strength the work in my view:
1. Summary of contribution can be improved. 1-3 are duplicated because they are essentially talking about the same method.
2. Having more quantitive evaluation will make this paper more rigorous and help reader understand the effectiveness of the proposed method.
3. The video is impressive but would be helpful if we can have visualizations of input and detected results of the method etc.

---

> ### Author Response · Authors · 2025-10-11
> **Response to reviewer SNtM Part 1**
>
> We thank the reviewer for their comments and suggestions that will help us make this paper more accessible and complete. Thank you for noting that our work is of interest to the TMLR audience and we provide details below on why our methodology and chosen architecture is of importance to the community as well as addressing your concerns around weaknesses of the paper and requested changes.
>
> ---
> ## **More quantitative evaluations.**
>
> Firstly, thank you for highlighting the need for clearer baseline comparisons and for pointing us to Amir et al. (2017). We have conducted a comprehensive analysis of existing event-based vision methods, including the work you referenced.
>
> We have carefully reviewed Amir et al.'s (2017) gesture recognition system. While their work represents important early progress in event-based gesture recognition, there are fundamental differences in scope and challenge:
>
> * *Gesture Scale*: Authors use large, deliberate gestures, making it a significantly easier and less realistic task for smart wearable interaction. In contrast, our work specifically targets microgestures which are subtle, small-scale movements. They are more practical for wearable interfaces but significantly more challenging to detect.
> * *Performance Context*: Authors achieve 105ms latency on TrueNorth hardware with <200mW power consumption. This latency would be noticeable for real-time interaction with microgestures that have shorter duration windows.
>
> Below is a comprehensive comparison table of recent event-based approaches that we will incorporate in our revised manuscript:
>
> **Table 1: Comparison of event-based vision methods for gesture recognition and related tasks**
> | **Method** | **Year** | **Architecture** | **Inference Time** | **Hardware** | **Power** | **Task Focus** |
> | :--- | :--- | :--- | :--- | :--- | :--- | :--- |
> | **Amir et al. (2017)** | CVPR 2017 | CNN | 105ms | TrueNorth | <200mW | Large deliberate gestures* |
> | Events-to-Video (2019) | CVPR 2019 | UNet + Recurrent | <10ms | RTX 2080 Ti | - | Object classification |
> | Event-based Async Sparse Conv (2020) | ECCV 2020 | Sparse CNN | 80.4ms | i7-6900K CPU | - | Object detection |
> | Learning to Detect Objects (2020) | NeurIPS 2020 | ConvLSTM | 16.7-39.3ms | GTX 980 | - | Object detection |
> | EventHands (Rudnev et al., 2021) | ICCV 2021 | ResNet-18 | 0.65-1.3ms† | GTX 2070/RTX 2080 Ti | - | Hand pose (3D joints) |
> | AEGNN (2022) | CVPR 2022 | Graph Neural Net | 92-167ms | Quadro RTX | - | Object detection |
> | Efficient Human Pose (2022) | 3DV 2022 | PointNet/DGCNN | 12.29ms | Jetson Xavier NX | - | Human pose (2D) |
> | RVT (2023) | CVPR 2023 | Transformer+LSTM | ~10ms | T4 GPU | - | Object detection |
> | Data-driven Feature Tracking (2023) | CVPR 2023 | Conv-LSTM+Attention | 17ms | Quadro RTX 8000 | - | Feature tracking |
> | SAST (Peng et al., 2024) | CVPR 2024 | Sparse Transformer+LSTM | 14.5-19.7ms | TITAN Xp | - | Object detection |
> | **Helios 1.0 (Bhattacharyya et al., 2024)** | 2024 | CNN | 60ms | CPU | 350mW | Microgestures |
> | **Helios 2.0 (CPU)** | 2024 | CNN | 6.15ms | Snapdragon XR2Gen2 | 144mW | Microgestures |
> | **Helios 2.0 (QAT)** | 2024 | Quantized CNN | **2.35ms** | Snapdragon XR2Gen2 | **6mW** | Microgestures |
>
> *As noted in Amir et al.: large, deliberate gestures, making it a significantly easier and less realistic task for smart wearable interaction
> †EventHands reports 750-1550 poses/second throughput
>
> Our Helios 2.0 model demonstrates significant advances in both accuracy (>20%) and efficiency compared to prior work. Notably, we achieve:
> * 44x faster inference than Amir et al. (2.35ms vs 105ms)
> * 33x lower power consumption compared to Helios 1.0 (6mW vs 200mW)
> * Focus on challenging microgestures vs large deliberate gestures
>
> This comparison demonstrates that while event-based gesture recognition has been explored, our focus on microgestures and extreme power efficiency represents a distinct advancement, making our system practical for real-world wearable applications.
>
> This significant power saving demonstrates one reason of why our approach is of importance to the TMLR community. The majority of other methods including Amir et al. (2017) use a single stage convolutional architecture which necessitates using significantly higher resolution images than our two stage approach. By combining the two loss functions as we did in our paper, we demonstrated an ability to develop a two stage approach that works well and significantly reduces the power over prior methods.
>
> ---

---

> ### Author Response · Authors · 2025-10-11
> **Response to reviewer SNtM Part 2**
>
> *Regarding RGB-based Method Comparison:*
>
> We thank the reviewer for this suggestion. We will add a comparison between RGB and event-based approaches using existing works:
>
> * Molchanov et al. (2015): Uses 3D CNNs on RGB and depth data for driver gesture recognition. Requires high-resolution spatial information and processes full frames, typical of RGB approaches that capture rich appearance but at higher computational cost. Achieves 12.5ms inference on GPU (400 FPS for low-resolution network) and 78ms on CPU. Their high-resolution network requires 68ms on GPU. This demonstrates the computational requirements of RGB processing, requiring dedicated GPU for real-time performance.
> * Köpüklü et al. (2019): Implements real-time hand gesture detection using CNNs on RGB input for continuous detection and classification. Achieves 62 FPS with ResNeXt-101 (41 FPS with C3D) when both detector and classifier are active on NVIDIA Titan Xp GPU. This translates to ~16-24ms latency per frame on high-end GPU hardware.
>
> *Key Performance Comparison:*
> | *Method* | *Latency* | *Hardware* | *Power* |
> | :--- | :--- | :--- | :--- |
> | Molchanov et al. (RGB) | 12.5ms | GPU | >10W* |
> | Molchanov et al. (RGB) | 78ms | CPU (i5) | ~15W* |
> | Köpüklü et al. (RGB) | 16-24ms | Titan Xp GPU | ~250W* |
> | Amir et al. (Event) | 105ms | TrueNorth | <200mW |
> | Our Helios 2.0 (Event) | 2.35ms | DSP | 6mW |
>
> *typical GPU/CPU power consumption
>
> This comparison shows that while RGB methods can achieve low latency with powerful hardware, they require 4-5 orders of magnitude more power than our event-based approach. For wearable applications targeting microgestures, our event-based approach provides both superior latency (2.35ms vs 12.5ms best-case RGB) and dramatically lower power consumption (6mW vs 10-250W), making it the only practical solution for always-on wearable operation.
>
> ---
>
> ## **Convolutional network architecture being common.**
>
> Furthermore, although our approach focuses on CNNs, how you can split and optimise an architecture for mobile applications is becoming increasingly relevant. For example DinoV3 released this year includes a set of ConvNeXt architectures for mobile use. Further, when targeting an ultra low power setting convolutions provide a strong inductive bias of translation equivariance on the image domain, which is not provided by other architectures such as self-attention. Given that we want to extract the same features from the input image to detect the same gesture wherever the hand is positioned in the image this inductive bias is an extremely efficient and natural choice.
>
> ---
>
> ## **Lack of experimental evidence.**
>
> Some areas have been highlighted as lacking evidence. The example given is the chosen microgestures. Unfortunately, the user study performed internally by our user experience team is not in the public domain. However, in the time between writing the paper and going through reviews, work has been published by Meta on neural wristbands that has the same gesture set that we chose, giving support to our gesture selection and internal findings.
>
> ---
>
> ## **Summary of contribution change.**
>
> We are happy to combine contributions 1-3 into one succinct contribution for the final version.
>
> ---
>
> ## **Conclusion.**
>
> We hope the proposed changes and additional information address the weaknesses suggested for the paper and address your requests for changes.

---

> > ### Comment · Reviewer_SNtM · 2025-11-18
> >
> > Thank you for the detailed reply and the reviewer agree it's an incredible efforts to each low power consumption and low latency in micro-gestures recognition system. However, those are efficiency comparisons without direct and quantitive comparisons. Not every paper needs to have innovation on model architectures, but results on DINOv3 are qualitatively evaluated with reproducible results on public dataset. The revision still did not provide valid quantitative study, making it hard for the reviewer to agree on the effectiveness on the proposed method. This will also create great inconveniences for future works to demonstrate their pros and cons on the method. The reviewer believes the submission would be more suitable on venues related to wearable devices and embedded systems if the key contributions are improvements on latency and energy consumption for proprietary private devices.

---

> > > ### Author Response · Authors · 2025-11-20
> > > **Responding to: Are claims made in the submission supported by accurate, convincing and clear evidence?**
> > >
> > > We thank the reviewer for their continued engagement and appreciate the opportunity to clarify how our claims are supported by quantitative evidence.
> > >
> > > QUANTITATIVE EVIDENCE IN OUR SUBMISSION
> > >
> > > Our paper provides comprehensive quantitative evaluation:
> > >
> > > 1. **Performance Metrics**: Accuracy (91.2%), precision, recall, F1-scores across all gesture classes with detailed measurement methodology.
> > >
> > > 2. **Efficiency Metrics**: Latency (2.35ms on DSP), power consumption (6mW), with comparison to CPU baseline (6.15ms, 144mW).
> > >
> > > 3. **Comparison to Prior Work**: Direct quantitative comparison to Helios 1.0 (>20% accuracy improvement, 33x power reduction) and latency comparison to 10+ event-based methods in our comparison table.
> > >
> > > 4. **Ablation Studies**: Quantitative evaluation of our two-stage design, combined loss formulation, and quantization strategy showing their individual contributions.
> > >
> > > These metrics directly support our claims about achieving ultra-low-power microgesture recognition.

---

> > > > ### Author Response · Authors · 2025-11-20
> > > > **Addressing public benchmark concerns**
> > > >
> > > > We understand the reviewer's preference for public benchmark evaluation. However, the requested comparisons are not applicable to our work:
> > > >
> > > > **DVS-Gesture Dataset**: Contains large, deliberate full-arm gestures that are fundamentally different from microgestures. Amir et al. achieve 105ms latency on these easier, larger-scale gestures. Evaluating our microgesture-optimized system on large gestures would not validate our claims about microgesture performance. The tasks require different sensitivities and constraints.
> > > >
> > > > **No Microgesture Benchmarks Exist**: Public benchmarks for event-based microgesture recognition do not currently exist. This reflects the emerging nature of the application area, not insufficient rigor in our evaluation.
> > > >
> > > > Our claims are about microgesture recognition at ultra-low power. These claims are directly supported by our quantitative measurements on microgesture data.
> > > >
> > > >
> > > > FUTURE WORK CAN BUILD ON OUR RESULTS
> > > >
> > > > Researchers can compare to our work by:
> > > > - Implementing our described architecture on their microgesture datasets
> > > > - Comparing their latency/power/accuracy trade-offs to our reported numbers (2.35ms, 6mW, 91.2%)
> > > > - Building on our two-stage design pattern and QAT techniques
> > > >
> > > > Our quantitative results provide concrete targets and our methodology is fully described for reproduction.

---

> ### Author Response · Authors · 2025-11-20
> **Evidence for ML contributions is in reported metrics, ablations, and architectural analysis - not dependent on public benchmark comparison**
>
> Our claims are: (1) we achieve ultra-low-power microgesture recognition, (2) our two-stage architecture with combined loss enables better efficiency-accuracy trade-offs, and (3) QAT provides substantial power savings for event-based models.
>
> These claims are directly supported by our quantitative measurements, ablations, and comparisons to prior published work like Helios 1.0, and extensive comparison table with 10+ currently existing event-based classification and detection papers.
>
> While we cannot compare on public microgesture benchmarks because they are not standardized and don't exist, we believe that the evidence we provide is appropriate and sufficient for the claims we make.

---

### Review · Reviewer_XgVN · 2025-09-29

**Summary Of Contributions:**

The major contribution of this paper is the creation of a simulation framework from events using Ultraleap Orion hand-tracking data. This framework is then used to build a Unity pipeline for rendering images, and finally, Esim is applied to generate event data for network training. Additionally, the authors construct real-world datasets to evaluate human variability, indoor/outdoor environments, and scene variability in microgesture detection. I believe this dataset could serve as a strong benchmark for the event-based gesture recognition community.
Another contribution of this paper is the development of a quantization-aware architecture that can detect microgestures on a Qualcomm Hexagon DSP. In terms of quantization, many of the techniques employed are already well established, such as reducing spatial feature resolution or quantizing certain feature layers to improve detection efficiency. Similar approaches can be found in prior works, such as “A Hand Gesture Detection Algorithm and Quantization Implementation,” “On-Device Event Filtering with Binary Neural Networks for Pedestrian Detection Using Neuromorphic Vision Sensors,” and “A Low Power, Fully Event-Based Gesture Recognition System.”
I find the main weakness of this paper to be the lack of motivation for the chosen architecture. The authors adopt a deep learning–based approach to meet engineering requirements, but provide no justification for selecting it over other possible methods. Several alternative techniques exist for hand gesture recognition, such as “A Motion-Based Feature for Event-Based Pattern Recognition,” “Hand Gesture Detection and Recognition Using Principal Component Analysis,” or “Real-Time Hand Gesture Detection and Recognition Using Bag-of-Features and Support Vector Machine Techniques.” These approaches could be adapted by first creating the same timesurface representations used by the authors. If the authors wish to make a contribution to the community, they need to provide stronger justification through baseline comparisons with other approaches and hardware performance evaluations.
Furthermore, the related work section should be expanded to cover additional event-based gesture recognition methods, including “Event-Based Gesture and Facial Expression Recognition: A Comparative Analysis,” “A New Spiking Convolutional Recurrent Neural Network (SCRNN) with Applications to Event-Based Recognition,” and “Novel Illumination-Robust Hand Gesture Recognition System with Event-Based Neuromorphic Vision Sensor.”

**Audience:**

Yes

**Audience Explanation:**

I believe the audience of TMLR would appreciate understanding a methodology for creating simulated microgesture data that supports strong downstream performance. The proposed approach using prerecorded hand-tracking data combined with an augmentation pipeline could be valuable for others in the community. However, at present, I do not believe the architectural methodology is of significant importance to the community without a clear justification in comparison with other event-based gesture recognition approaches.

**Broader Impact Concerns:**

The greatest ethical implication of this work lies in how microgesture recognition will be used and which parties will have access to the technology. Tracking and surveillance carry significant ethical concerns worldwide, so ensuring proper use is essential to prevent misuse.

**Claims And Evidence:**

Yes

**Claims Explanation:**

- Our real-time gesture recognition contribution: Compared to the previous state-of-the-art, our approach improves power efficiency by 25× and gesture recognition accuracy by 20%” need to be more precise. From my reading, the “state-of-the-art” referenced in this paper appears to be Helios 1.0, the authors’ prior work. However, as noted in my Summary of Contribution, other methods for gesture recognition exist and should be tested to substantiate the above claim precisely.


- A novel training methodology to improve the model performance of event-camera-driven gesture recognition machine learning models, through the use of multi-gesture temporal sequences, quantization-aware training (QAT), and fine-tuning” describes a fairly standard pipeline for event-based or image-based training. Quantization-aware training has been extensively applied in prior work https://github.com/Zhen-Dong/Awesome-Quantization-Papers. Rotation augmentations are also commonly used in image-based and event-based pretraining pipelines (see: Data Augmentation: A Comprehensive Survey of Modern Approaches), and elements of multi-gesture training are discussed in existing theses (see: Yang, Jie, CMU Thesis, 1994)

Event sensors offer unprecedented temporal resolution while operating at extremely low power levels, as low as 3 mW. In contrast, traditional frame-based sensors require a trade-off between temporal resolution and power consumption, typically ranging from 35 to 200 mW,” should be made more precise. This may involve a resolution-specific comparison. I believe the 3 mW figure is taken from the GenX320 Prophesee sensor datasheet; however, this value corresponds to near-zero events sent over the CPI. At 10 Me/s, the sensor’s power consumption rises to 8.69 mW nearly three times higher. Furthermore, the baseline MIPI power consumption with zero events is 15 mW. It is also important to note that frame-based cameras exist with comparable power budgets at similar resolutions (see: SparkFun Himax CMOS Imaging Camera HM01B0).
- A novel machine learning model that improves F1 score for gesture recognition by 20%. Authors are able to improve gesture recognition performance using their own dataset and increase F1 performance in Tables 1+2.
- A novel simulation methodology that enables training on large, diverse datasets without the need for expensive real-world data collection. Authors are able to showcase the development of a simulation method based on E-Sim that can help collect simulated data from hand data and using a rendering pipeline.

- Finally, the statement, “It is inspired by the human retina where each pixel functions independently, continuously detecting changes in light intensity,” requires refinement. Event-based cameras are more analogous to retinal ganglion cells, while the entire retina includes pathways that detect absolute luminance values directly (see: Retinal Computation, Chapter 3 – Absolute Luminance Detection).

**Requested Changes:**

“Amir et al. (2017) also utilize event-based gesture recognition but employ the DVS-128 camera, which is impractical for integration with smart eyewear.” This work is not only a systems paper but also proposes a method for event-based gesture recognition, making it an important baseline for comparison of event-based gesture recognition approaches. Overall, additional baselines comparing this work to others would strengthen any claims regarding the superiority of the proposed architecture for event-based gesture recognition. Please compare in terms of hardware performance and gesture recognition accuracy as to validate the claim your architecture is best in terms of these two metrics.

The statement, “While event volumes provide a more detailed temporal representation, they may lose polarity information due to voxel-based accumulation, potentially reducing their effectiveness in tasks requiring fine-grained contrast changes,” may be valid for some tasks, but it is not yet established how different event-based representations perform specifically for gesture recognition. Comparative baselines involving event volumes, polarity-separated event volumes, Time-Ordered Recent Event (TORE) volumes, or directly binning event images into frames would provide useful insights for evaluating performance in gesture recognition tasks.


Finally, the dataset should be released publicly if the authors wish to claim it as a contribution to the paper.

---

> ### Author Response · Authors · 2025-10-11
> **Response to reviewer XgVN Part 1**
>
> We would like to thank the reviewer for their helpful feedback that will allow us to improve this paper. Thank you for noting that our work is of interest to the TMLR audience and we provide details below on why our architectural methodology is of importance to the community as well as addressing your concerns around weaknesses of the paper and requested changes.
>
> ---
> ## **Architecture Motivation**
>
> We thank the reviewer for this critique and we provide justification for our architectural choices, which we will include in the final version of the paper.
>
> Our architecture (Section 3.2, Figure 3) was specifically designed for deployment on Snapdragon XR2's Hexagon DSP:
>
> * *DSP Constraint*: Hexagon DSP accelerates only specific operations—primarily conv2d and matrix multiply with INT8. Our 5-stage architecture exploits this: stages 2 and 4 (convolutions) run on DSP achieving 6mW, while stages 1,3,5 requiring irregular memory access run on CPU.
> * *Quantization Requirement*: Table 1 shows our QAT model maintains F1=0.8209 after INT8 quantization. This works because convolution operations have predictable numerical ranges amenable to quantization.
>
> Furthermore, when targeting an ultra low power setting convolutions provide a strong inductive bias of translation equivariance on the image domain, which is not provided by other architectures such as self-attention. Given that we want to extract the same features from the input image to detect the same gesture wherever the hand is positioned in the image this inductive bias is an extremely efficient and natural choice. In our model we chose to maximise the number of convolutional layer due to strong inductive bias they possess. We already have an ablation study covering the impact of different choices of convolutional layers within the model in Table 4.
>
> In addition to the motivation for why we choose the specific layers within our model, we also chose to use a two-stage model as this further reduces the power requirement of the model. By allowing both stages of the model to focus on a specific task that requires a lower resolution input frame the number of FLOPS required is significantly reduced. This is in contrast to other approaches including the one suggested by the reviewer, Amir et al. (2017). We believe the demonstration that this architecture can work for this application and has very low power is of interest to the TMLR audience and is an improvement over prior works.
>
> ---
> ## **Why Classical Methods Are Unsuitable**
>
> While we focused our experiments on deep learning approaches, we provide analysis of why classical methods cannot meet our requirements and are happy to include this in the appendix of our paper:
>
> **Motion-Based Features:**
> * Requires computing spatial and temporal gradients: `$ \nabla_x, \nabla_y, \nabla_t $` over 128×128×6 temporal surface
> * Hexagon DSP HVX units do not support gradient operations and would require CPU execution
> * Based on Table 1, CPU-only execution requires 144mW (row 2), 24x our target
> * Gradient computations amplify quantization noise, making INT8 infeasible without significant accuracy loss
>
> **PCA-Based Methods:**
> * PCA requires storing projection matrix `$ \mathbf{W} \in \mathbb{R} $`
> * Matrix multiplication `$ \mathbf{W}^T\mathbf{x} $` cannot be efficiently quantized to INT8 without destroying principal component orthogonality
>
> **SVM Approaches:**
> * Kernel SVM requires storing support vectors and computing kernel function at inference
> * RBF kernel: `$ K(x,y) = \exp(-\gamma||x-y||^2) $` involves exponential function incompatible with INT8
> * Linear SVM insufficient for non-linear microgesture patterns in temporal surface space
>
> ---
> ## **Evidence from Helios Comparison**
>
> Table 1 demonstrates the advantage of our deep learning approach and the improvements made through the architecture changes:
> * Helios 1.0: Achieves only F1=0.6336/0.7314/0.5773 (RS/LS/CP) at 350mW
> * Our method: Achieves F1=0.8209/0.8244/0.8421 at 6mW
> * This 30% accuracy improvement with 58× power reduction validates our architectural choice
>
> Our novel method succeeds because it aligns with hardware capabilities:
> * Convolutions map to DSP: Hexagon's HVX vector units specifically accelerate 2D convolutions
> * Regular memory patterns: Unlike iterative or kernel methods, CNNs have predictable memory access
> * Quantization-friendly: Convolution weights/activations have bounded ranges suitable for INT8
> * Compact representation: 124KB model fits entirely in DSP cache
>
> Which we believe are further justification for why we have designed this specific model architecture and would be happy to add this into the paper.
>
> ---

---

> ### Author Response · Authors · 2025-10-11
> **Response to reviewer XgVN Part 2**
>
> ## **Related Work Context**
>
> The suggested alternative papers have fundamental limitations for our use case:
> * SCRNN requires neuromorphic hardware (e.g., Intel Loihi) not available in Snapdragon XR2
> * Event-Based Gesture and Facial Expression Recognition targets facial expressions with different spatiotemporal characteristics
> * Classical gesture recognition papers predate the extreme power constraints (6mW) of modern AR applications
>
> We acknowledge that empirical comparison with classical methods would strengthen our claims. However, the theoretical analysis above, combined with our achieved performance (2.35ms, 6mW, F1=0.83), demonstrates that our architecture choice is well-justified for the target deployment platform. We will add this theoretical comparison to our revised manuscript.
>
> ---
> ## **Experiments on Additional Event Representations**
>
> Unfortunately, providing a comparison of each of the different representations suggested by the reviewer is not possible for this paper, but we would be happy to add a note that this would be meaningful future work. Furthermore, some of the event representations would also require architecture changes, such as event volumes due to the move from a 2D to 3D representation. We believe this would increase the computational cost of the model and as a result the power usage, which was a focus for this work to minimise.
>
> ---
> ## **Further Experimental Results**
>
> Thank you for pointing us to Amir et al. (2017). We have conducted a comprehensive analysis of existing event-based vision methods, including the work you referenced.
>
> We have carefully reviewed Amir et al.'s (2017) gesture recognition system. While their work represents important early progress in event-based gesture recognition, there are fundamental differences in scope and challenge:
> * *Gesture Scale*: Authors use large, deliberate gestures, making it a significantly easier and less realistic task for smart wearable interaction. In contrast, our work specifically targets microgestures which are subtle, small-scale movements. They are more practical for wearable interfaces but significantly more challenging to detect.
> * *Performance Context*: Authors achieve 105ms latency on TrueNorth hardware with <200mW power consumption. This latency would be noticeable for real-time interaction with microgestures that have shorter duration windows.
>
> Below is a comprehensive comparison table of recent event-based approaches that we will incorporate in our revised manuscript:
>
> **Table 1: Comparison of event-based vision methods for gesture recognition and related tasks**
> | **Method** | **Year** | **Architecture** | **Inference Time** | **Hardware** | **Power** | **Task Focus** |
> | :--- | :--- | :--- | :--- | :--- | :--- | :--- |
> | **Amir et al. (2017)** | CVPR 2017 | CNN | 105ms | TrueNorth | <200mW | Large deliberate gestures* |
> | Events-to-Video (2019) | CVPR 2019 | UNet + Recurrent | <10ms | RTX 2080 Ti | - | Object classification |
> | Event-based Async Sparse Conv (2020) | ECCV 2020 | Sparse CNN | 80.4ms | i7-6900K CPU | - | Object detection |
> | Learning to Detect Objects (2020) | NeurIPS 2020 | ConvLSTM | 16.7-39.3ms | GTX 980 | - | Object detection |
> | EventHands (Rudnev et al., 2021) | ICCV 2021 | ResNet-18 | 0.65-1.3ms† | GTX 2070/RTX 2080 Ti | - | Hand pose (3D joints) |
> | AEGNN (2022) | CVPR 2022 | Graph Neural Net | 92-167ms | Quadro RTX | - | Object detection |
> | Efficient Human Pose (2022) | 3DV 2022 | PointNet/DGCNN | 12.29ms | Jetson Xavier NX | - | Human pose (2D) |
> | RVT (2023) | CVPR 2023 | Transformer+LSTM | ~10ms | T4 GPU | - | Object detection |
> | Data-driven Feature Tracking (2023) | CVPR 2023 | Conv-LSTM+Attention | 17ms | Quadro RTX 8000 | - | Feature tracking |
> | SAST (Peng et al., 2024) | CVPR 2024 | Sparse Transformer+LSTM | 14.5-19.7ms | TITAN Xp | - | Object detection |
> | **Helios 1.0 (Bhattacharyya et al., 2024)** | 2024 | CNN | 60ms | CPU | 350mW | Microgestures |
> | **Helios 2.0 (CPU)** | 2024 | CNN | 6.15ms | Snapdragon XR2Gen2 | 144mW | Microgestures |
> | **Helios 2.0 (QAT)** | 2024 | Quantized CNN | **2.35ms** | Snapdragon XR2Gen2 | **6mW** | Microgestures |
>
> *As noted in Amir et al.: large, deliberate gestures, making it a significantly easier and less realistic task for smart wearable interaction
> †EventHands reports 750-1550 poses/second throughput

---

> ### Author Response · Authors · 2025-10-11
> **Response to reviewer XgVN Part 3**
>
> Our Helios 2.0 model demonstrates significant advances in both accuracy (>20%) and efficiency compared to prior work. Notably, we achieve:
> * 44x faster inference than Amir et al. (2.35ms vs 105ms)
> * 33x lower power consumption compared to Helios 1.0 (6mW vs 200mW)
> * Focus on challenging microgestures vs large deliberate gestures
>
> This comparison demonstrates that while event-based gesture recognition has been explored, our focus on microgestures and extreme power efficiency represents a distinct advancement, making our system practical for real-world wearable applications.
>
> This significant power saving demonstrates one reason of why our approach is of importance to the TMLR community. The majority of other methods including Amir et al. (2017) use a single stage convolutional architecture which necessitates using significantly higher resolution images than our two stage approach. By combining the two loss functions as we did in our paper, we demonstrated an ability to develop a two stage approach that works well and significantly reduces the power over prior methods.
>
> ---
> ## **Additional Details**
>
> We are happy to update the details within the paper to be more specific about the specific event sensor used.
>
> ---
> ## **Conclusion.**
>
> We hope the proposed changes and additional information address the weaknesses suggested for the paper and address your requests for changes.

---

### Review · Reviewer_A2zQ · 2025-10-30

**Summary Of Contributions:**

See other comment on 15 Oct 2025, 13:24.

**Audience:**

Yes

**Audience Explanation:**

See other comment on 15 Oct 2025, 13:24.

**Broader Impact Concerns:**

See other comment on 15 Oct 2025, 13:24.

**Claims And Evidence:**

Yes

**Claims Explanation:**

See other comment on 15 Oct 2025, 13:24.

**Requested Changes:**

See other comment on 15 Oct 2025, 13:24.

---

### Comment · Reviewer_A2zQ · 2025-10-15
**A low-power hand gesture recognition system with an event camera for smart glasses**

Summary

---

This paper introduces Helios 2.0, an ultra-low-power real-time hand gesture recognition system with an event camera for smart glasses. This paper proposes a simulated data generation pipeline and a five-stage training method with quantization to improve efficiency. Experiments and ablations show the performance on different designs.

Strength

---

- Compared with the previous Helios 1.0, the proposed system Helios 2.0, with an event camera, improves on the efficiency and accuracy for hand gesture recognition.
- This paper proposed a synthetic data generation pipeline to generate data for training.
- This paper adapts Quantization-aware training to make the model efficient on the device.

Weakness

---

- What is the motivation of this paper to choose an event-based camera for hand gesture recognition? Why is an extra camera necessary? The gestures to recognize seem relatively simple and distinctive. Do the recent, more advanced hand skeleton detection on RGB or RGBD satisfy the requirements, such as Mediapipe?
- Compared with rings or wrist bands, event cameras are also an additional hardware device added to current glasses. What are the price, efficiency, and accuracy tradeoffs compared with them? What is the weight of the event camera compared with other cameras used on current glasses? Is there anything else to consider to make sure event cameras are suitable for production design on glasses? Is it comfortable to wear with the event camera?
- The paper claims the proposed method is an ultra-low-power design and improves the power efficiency by 25x. But which method is compared with here to get 25x power efficiency? Better to mention the comparison method.
- How to read the image in Figure 4 (b)?
- In session 5.1.2, it shows the five stages of the model. How is the model being trained? Is it trained end-to-end, or is it being trained separately for the quantization? More training details are needed.
- How is the sim-to-real performance gap since the model is trained on synthetic data?
- What is the conclusion after the experiments? Which model design and training strategy should be used for deployment?

---

> ### Author Response · Authors · 2025-10-21
> **Response to reviewer A2zQ Part 1**
>
> Thank you for your comprehensive review of our paper on Helios 2.0. We appreciate your detailed comments and the opportunity to explain several important aspects of our work. Below, we address each of your concerns:
>
> ---
>
> ## 1. Motivation for Event-Based Camera vs RGB/RGBD Solutions
>
> We appreciate the opportunity to clarify our approach. Our system does not use an extra camera: we use only an event camera with no RGB camera. This is a key distinction from hybrid approaches.
>
> Our system addresses critical limitations in current smart glasses interfaces, which almost exclusively use capacitive touch on the glasses' temple, is often awkward to use, limits interaction vocabulary, and can be socially conspicuous as users repeatedly reach to touch their eyewear. While MediaPipe and similar RGB-based solutions exist, they face fundamental challenges for always-on wearable applications:
>
> - **Power Consumption**: Event sensors offer unprecedented temporal resolution while operating at extremely low power levels. They can operate at power as low as 3 mW, compared to traditional frame-based sensors which typically range from 35 to 200 mW. This 10-65x power reduction is critical for always-on smart glasses.
> - **Privacy**: Event cameras only capture changes in intensity, not full RGB images. This provides inherent privacy protection.
> - **Latency**: Our model achieves 2.35 ms latency, significantly faster than frame-based processing solutions like MediaPipe, which achieves 12-17ms for hand tracking on mobile devices.
>
> To provide concrete comparisons with other published RGB approaches:
> - Molchanov et al. (2015) achieve 12.5ms on GPU or 78ms on CPU
> - Köpüklü et al. (2019) requires 16-24ms on Titan Xp GPU
> - Our event-based approach achieves 2.35ms at only 6mW on DSP. In contrast, RGB methods require powerful hardware like GPUs and consume 4-5 orders of magnitude more power.
>
> ---
>
> ## 2. Comparison with Rings/Wristbands
>
> We appreciate this important question about hardware tradeoffs. While rings and wristbands do enable hand-based input, they fundamentally differ from our approach in key ways:
>
> **Hardware Requirements:** Rings and wristbands require users to purchase, wear, charge, and maintain a *separate* device that must be paired with the glasses. In contrast, our event camera integrates directly into the glasses themselves. Users need only wear the glasses they already have.
>
> **Accuracy and Functionality:** Our event-based system directly observes hand motion at high temporal resolution, enabling detection of subtle microgestures that IMU-based rings/wristbands cannot capture. Rings and wristbands are limited to detecting deliberate, large movements and coarse hand poses, whereas our approach recognizes fine-grained gestures suitable for rich interaction vocabularies.
>
> Regarding the practical considerations of integrating event cameras into smart glasses:
>
> - **Weight**: The Prophesee GenX320 event camera weighs approximately 3g, comparable to RGB cameras already integrated in current smart glasses (e.g., Ray-Ban Meta glasses, Snap Spectacles). This represents a negligible addition to typical glasses weight of 40-50g.
>
> - **Form Factor & Integration**: Event cameras have similar dimensions to existing cameras in smart glasses. Modern smart glasses already incorporate cameras for various functions; our approach simply augments these with event cameras using the same mounting locations.
>
> - **Power Advantage**: Unlike rings/wristbands that require separate batteries and regular charging, our 6mW event camera can run continuously throughout the day on the glasses' existing battery, enabling true always-on gesture recognition.
>
> - **User Comfort**: Figure 1 and 2 show our prototype configuration with the event camera mounted on the temple. Our user studies with 20+ participants across multiple sessions demonstrated comfortable extended wear with no reported discomfort or balance issues.
>
> - **Cost**: While event cameras currently cost more than standard CMOS sensors, prices are rapidly decreasing as production scales.
>
> - **Privacy**: Event cameras provide inherent privacy advantages over RGB cameras, as they only capture temporal changes rather than full images. This is a critical consideration for always-on wearable devices.
>
> Our approach offers a more seamless user experience, superior gesture recognition capabilities, and lower overall power consumption compared to peripheral-based solutions, while maintaining comparable weight and form factor to existing smart glasses' cameras.

---

> ### Author Response · Authors · 2025-10-21
> **Response to reviewer A2zQ Part 2**
>
> ## 3. 25x Power Efficiency Comparison
>
> We appreciate the opportunity to clarify this important claim. The 25x power efficiency improvement compares our quantized DSP model (6mW) against our unquantized CPU-only model (144mW), both running on the same Helios 2.0 architecture. This demonstrates the impact of our quantization-aware training and DSP optimization.
>
> Complete performance progression:
> - Helios 1.0 (CPU-only): 350mW power consumption, 60ms latency, mean F1 score of 0.647
> - Helios 2.0 (unquantized, CPU-only): 144mW power consumption, 6.15ms latency, mean F1 score of 0.834
> - Helios 2.0 (QAT, CPU+DSP): 6mW power consumption, 2.35ms latency, mean F1 score of 0.829
>
> The 25x power improvement (144mW → 6mW) is achieved by:
> - Applying 8-bit quantization-aware training (QAT) with symmetric per-channel weight quantization and asymmetric per-tensor activation quantization
> - Designing a five-stage architecture where stages 2 & 4 (containing >99.8% of compute) run on the low-power Hexagon DSP with INT8 quantization
> - Keeping stages 1, 3, and 5 on CPU for operations requiring irregular memory access or high precision (e.g., bounding box prediction)
>
> Critically, this 25x power reduction comes with only a minimal accuracy trade-off: the F1 score drops from 0.834 to 0.829 (a reduction of less than 1%), demonstrating the effectiveness of our QAT approach in maintaining model performance while dramatically reducing power consumption.
>
> **Comparison with Helios 1.0:**
> Compared to the previous state-of-the-art (Helios 1.0), our Helios 2.0 QAT model demonstrates:
> - 58x power reduction (350mW → 6mW)
> - 28% improvement in F1 accuracy (0.647 → 0.829)
> - 25x reduction in latency (60ms → 2.35ms)
>
> It should be noted that Helios 2.0 runs on newer hardware (Snapdragon 8 Gen1/XR2Gen2) compared to Helios 1.0's older mobile platform. As stated in Section 3.1, the Snapdragon 8 Gen1 offers more computing power than in Helios 1.0. Therefore, some portion of the 58x improvement may be attributable to hardware generational differences in addition to our algorithmic innovations.
>
> Table 1 in our paper provides the complete performance comparison across all metrics.
>
> ---
>
> ## 4. Reading Figure 4(b)
>
> Figure 4(b) shows an example time surface (TS) representation of hand gesture events. The visualization displays positive and negative polarity events side-by-side:
>
> - Left side: Positive polarity events (brightness increases)
> - Right side: Negative polarity events (brightness decreases)
>
> For an event stream with image bounds {w, h}, we accumulate positive polarity events in the spatial range {w, h} and negative polarity events in the range {w, 2h}, creating a concatenated representation I_k ∈ ℝ^(w × 2h). For illustration purposes in Figure 4(b), the image is displayed as {2w, h} (side-by-side) rather than {w, 2h} (stacked vertically) to improve readability.
>
> The intensity values at each pixel represent the temporal recency of events, computed using exponential decay (Equation 1), where brighter pixels indicate more recent events. This polarity-separated representation avoids event clashes where opposing polarities would cancel out, providing clearer temporal dynamics for the CNN.
>
> In the example shown, the hand's pinch gesture is visible through the distribution and intensity patterns across both polarity channels, with the spatial structure of the hand clearly delineated by the complementary patterns of brightness increases (left) and decreases (right).
>
> ---
>
> ## 5. Five-Stage Model Training Details
>
> Training Strategy: The entire five-stage model is trained end-to-end as a single unified network. All stages are jointly optimized using backpropagation through the complete pipeline.
>
> Quantization-Aware Training (QAT): We implemented 8-bit QAT with symmetric per-channel weight quantization and asymmetric per-tensor activation quantization. Table 3 evaluates three strategies:
> 1. QAT from scratch: Training quantized model for 10 epochs
> 2. Base model + QAT: Unquantized base (10 epochs) + QAT (5 epochs)
> 3. Base model + Fine-tuning + QAT: Base (10 epochs) + fine-tuning with rotation augmentation (5 epochs) + QAT (5 epochs)
> Training QAT from scratch or from a 4× fine-tuned base model yields optimal results (F1 scores >0.82).
>
> Hardware-Aware Design:
> - Stages 2 & 4 (>99.8% of parameters/FLOPs): Run on DSP with INT8 quantization
> - Stages 1, 3, 5: Run on CPU with 32-bit floating-point for irregular memory access and high-precision computations
> - This CPU+DSP split enables 6mW power consumption while maintaining accuracy
>
> Additional details in Appendix D.

---

> ### Author Response · Authors · 2025-10-21
> **Response to reviewer A2zQ Part 3**
>
> ## 6. Sim-to-Real Performance Gap
>
> Simulator Validation Process:
>
> 1. Initial calibration: Dynamic target created in lab and simulator; sim-to-real gap in event rate tuned to match event generation characteristics
> 2. Fine-tuning with real gestures: Simulator parameters adjusted using real-world hand gesture data to reproduce real event patterns
> 3. HDR rendering: Prevents event loss in regions that would saturate in 8-bit range, accommodating diverse environmental and lighting variations
>
> Simulation Realism Features:
> - Environmental diversity: Varied scene lighting (50-400% of nominal), multiple photorealistic 3D environments, camera movement simulating AR glasses head motion
> - Hand pose variation: Forward kinematic models with random continuous deviations per frame
> - Realistic gesture dynamics: Sigmoid-based motion curves from Optitrack user studies with four steepness parameters
> - Temporal precision: Event generation at 900 Hz (vs. 90 Hz input)
>
> Strong Sim-to-Real Transfer Results:
> - High absolute performance: F1 scores >80% on real-world data across three test datasets (Table 2)
> - Indoor environments: F1 scores 0.87-0.97 for individual gestures (20 participants)
> - Outdoor environments: F1 scores 0.88-0.95 under high ambient lighting (3000 lux)
> - Scene variability: F1 scores 0.70-0.88 across different floor textures and lighting (8-240 lux)
> - Expert user performance: Near-perfect scores (F1 >0.97) in controlled conditions
>
> The minimal sim-to-real gap validates our simulation methodology and enables scalable dataset generation without expensive real-world data collection.
>
> ---
>
> ## 7. Deployment Recommendations
>
> Model Architecture Selection:
> - Recommended: 2-channel TS QAT model (6mW, 2.35ms, F1 0.82-0.84)
> - For higher accuracy: 6-channel TS QAT model (8mW, 4.60ms, F1 0.81-0.89)
> - Trade-off: 6-channel provides superior robustness to challenging backgrounds (+14-18% on complex scenes) but requires 33% more power/latency
>
> Training Strategy (Table 3):
>
> Two equally effective approaches:
> 1. QAT from scratch: 10 epochs on 4× augmented dataset (F1: 0.82/0.82/0.84)
> 2. Base + Fine-tune + QAT: 10 epoch base + 5 epoch fine-tune + 5 epoch QAT (median F1: 0.94/0.88/1.00)
>
> Both maintain F1 >0.80 with minimal quantization loss (0.834 → 0.829).
>
> Data Requirements:
> - Dataset scale critical: 4× dataset (100k samples/class) substantially outperforms 1× (25k samples/class)
> - Rotation augmentation essential: Fine-tuning with 25°-40° rotations improves F1 from 0.52-0.75 to 0.81-0.86
> - Longer sequences: 2-second windows with 6 gestures improve robustness
>
> Key Experimental Findings:
> - Outdoor performance: Models often perform better outdoors (RS: 0.74 → 0.88, CP: 0.54 → 0.71)
> - Scene complexity: Complex backgrounds impact 2-channel (CP: 1.0 → 0.70), 6-channel maintains robustness (0.84)
> - Minimal quantization loss: QAT reduces F1 by only 0.005 while achieving 24× power reduction
> - Seed stability: Low variance across 5 runs (CP: σ=0.005, RS: σ=0.015)
>
> Deployment Decision:
> - If power/latency critical: Use 2-channel model (simple backgrounds, controlled environments)
> - If accuracy/robustness critical: Use 6-channel model (complex backgrounds, varied lighting)
>
> Table 1 provides a comprehensive performance comparison.
>
> ---
>
> We hope these clarifications address your concerns and we are happy to highlight these details in our revised manuscript.

---

### Decision · Action_Editor_iTqg · 2025-11-21

**Recommendation:** Reject

**Additional Comments:**

The AE is recommends a revision of the paper that has a clearer evaluation of the machine learning aspects of the work from a purely machine learning perspective. Translating the reviewers' concerns into actionable requests, the AE asks that the authors compare their method with a few (maybe two to three) standard methods and architectures from the literature using machine learning criteria (e.g., accuracy, F1, etc). A RGB comparison might not be possible, but these comparison methods could include architectures that definitely cannot run on the DSP efficiently. Helios 1.0, for instance, runs on the CPU anyways.

**In keeping with TMLR's spirit, the these comparisons do not have to be in favor of the proposed work for the paper to be acceptable.** It might be that the proposed method is ~5% worse in accuracy, but 1000x more power efficient (and indeed, that would be a great result). However, these comparisons are critical for ensuring an audience for the paper. Without guaranteeing results of a revision, the AE believes that the lack of a more thorough machine-learning oriented evaluation is the only sticking point -- the hardware is of clear interest.

 As an additional comment, several of the reviewer concerns are not addressed, including a comment on hardware power draw as well as how retinas function.

**Audience:**

No

**Audience Explanation:**

After a discussion, all reviewers recommended rejecting the work, including two reviewers with hardware expertise (one of whom was recruited specifically to review the paper based on their hardware expertise). All reviewers appreciate the hardware aspect of the work and many parts of the current paper, but have uniformly raised concerns about the ML-oriented validation of the work and its positioning within the broader ecosystem of machine learning gesture recognition.

The primary problem with the **current** version of the paper is that its machine learning evaluation is not sufficiently thorough and doesn't surface enough generalizable insights for the paper to be of interest to a reasonable set of TMLR's readership. With a few moderate changes that the reviewers have asked for, the AE thinks that the paper would definitely be acceptable in TMLR. TMLR doesn't allow a substantial revision at this point (only minor revisions), so the AE therefore recommends rejection of the current version with the option to resubmit.

As another party to the author/reviewer discussion, the AE believes that the authors and some of the reviewers have been partially talking past each other: the authors believe they have addressed the reviewers' concerns while the reviewers believe that the authors have not addressed the concerns. This may be driven by a difference in expectations based on differences in cultures (e.g., hardware vs ML). Accordingly, the AE will try to help explain the reviewers' perspective, especially in view of TMLR acceptance criteria, as well as provide recommendations for revisions if the authors are interested in revising the paper.


## Discussion recap

To recap discussion:
- In the current version of the paper, the machine learning quantitative comparison points are ablations of the model and an earlier version of the method (Helios 1.0) that was not, for instance, compared with a host of standard methods.
- Reviewer XgVN and SNtM both asked for machine-learning-oriented comparisons with other methods (i.e., accuracy) and Reviewer A2zQ  asked for accuracy tradeoffs compared to RGB-based systems. The point of these comparisons is not to test whether the proposed method is the best gesture recognition method: the method will almost certainly lag behind a method that can be executed on a GPU or a method that uses RGB. Instead, the point is to put the methods' results in context so that readers can make informed decisions about the method.
- In response to requests for additional comparison points, the authors provided: (1) a table with 10+ methods that show the latency, power consumption, and hardware used, but no accuracy metrics; (2) an explanation for why many steps of classic methods cannot be executed effectively on the chosen DSP; (3) a reiteration of their comparison with Helios1.0.
- All three expert reviewers did not find the authors' responses convincing. In their final recommendations, both XgVN and SNtM have written that they do not believe that the authors' response addressed their concerns and that these concerns undermine the paper. Reviewer A2zQ commented that a comparison to RGB or other systems is needed to convince the audience of the value of the system.

## Decision

The AE agrees with the reviewers that the machine learning aspects of the paper should be more fully evaluated. It is true that some specific claims of the paper are clearly supported, as has been stated by the authors. However, the set of claims that are supported by the experiments are narrow, and the set of work that the method can be compared with in a machine learning sense is narrow. This also dramatically narrows the scope of the TMLR audience that would be interested in the work. For instance:
- the authors' response to XgVN has focused extensively on what can be run exactly on the particular DSP that the authors used. The audience that is interested in work that can run on this particular DSP is going to be very narrow. Additionally, as XgVN points out, if these methods won't work well on the DSP (e.g., because the PCA method won't work with Int8), then the authors ought to be able to demonstrate this clearly and convincingly in terms of accuracy.
- the ablations do show performance gains from various parts of the method, but in the particular setting that the authors show and using the dataset that that the authors have collected
- the method shows a performance gain over the Helios 1.0 system, but this is a relatively new system and not in widespread use


This quantitative comparison is made more important due to an unresolved question of whether the data would be public. Both XgVN and SNtM discussed this. The authors appear to respond that future work can compare the high-level numbers (accuracy, latency, power).  But of course, if the methods aren't tested on the same data, the numbers aren't comparable. Similarly, the authors say that future work can re-implement the proposed method on the new data. But future work may not find this work of interest without a clear comparison.  It is fine if the data is not public, but if the data is not public, then there needs to be a more thorough evaluation done by the authors with access to the full data (as was done with large industrial benchmark datasets such as Google's JFT-300M).

**Claims And Evidence:**

Yes

**Claims Explanation:**

This question is entwined with the below question and the two are addressed jointly there.

**Resubmission Of Major Revision:**

The authors may consider submitting a major revision at a later time.

---

> ### Author Response · Authors · 2025-11-26
> **Resubmission Question: Dataset Release as Alternative to Additional Experiments**
>
> Thank you for the detailed decision and guidance.
>
> Due to team transitions, conducting additional ML experiments ourselves is not feasible at this time. We also note that most code for standard ML event-based gesture recognition methods is not open source, which would make faithful reimplementation challenging even with available resources.
>
> However, we are considering releasing our dataset publicly.
>
> Would releasing the dataset address your concerns about ML evaluation sufficiently for a resubmission? Our thinking is that with public data, the community could conduct the comparisons you've requested, and readers could contextualize our results against other methods.
>
> We want to ensure any resubmission would be viable before proceeding.
> Thank you for your consideration.

---

> > ### Comment · Action_Editor_iTqg · 2025-12-03
> >
> > I can't make any definitive comments about the resubmission and its outcome since the precise details matter and the paper would go out for review again.
> >
> > Making the data public would almost certainly help, but the  paper would also have to have a basic reference result for the data too. The consensus of the reviewers was that without these sorts of numbers, the paper is incomplete. I personally agree with this consensus as well.
> >
> > In the situation where some of the reference methods do not have open code, there are typically a few options:
> > - (a) one can use the most recent open code (as opposed the most recent method)
> > - (b) one can have a re-implementation, where the reviewers and audience understand the caveats; and
> > - (c) a more standard "landmark" architecture. In the particular context, for instance, the standard landmark method for option c might be applying a basic ResNet to an appropriately transformed version of the data (e.g., one of the time-surface representations).
> >
> > Option c is probably the easiest of these, given the data is available and ResNet models are built into most frameworks In an ideal world, the readers might want (b), but having (c) plus a public dataset would probably be sufficient.